# Transcriptional heterogeneity and cell cycle regulation as central determinants of Primitive Endoderm priming

Marta Perera[1], Silas Boye Nissen[2†], Martin Proks[1], Sara Pozzi[1], Rita S Monteiro[1], Ala Trusina[2], Joshua M Brickman[1]*

[1]reNEW UCPH - The Novo Nordisk Foundation Center for Stem Cell Medicine, University of Copenhagen, Copenhagen, Denmark; [2]Niels Bohr Institute, University of Copenhagen, Copenhagen, Denmark

**Abstract** During embryonic development cells acquire identity as they proliferate, implying that an intrinsic facet of cell fate choice requires coupling lineage decisions to cell division. How is the cell cycle regulated to promote or suppress heterogeneity and differentiation? We explore this question combining time lapse imaging with single-cell RNA-seq in the contexts of self-renewal, priming, and differentiation of mouse embryonic stem cells (ESCs) towards the Primitive Endoderm (PrE) lineage. Since ESCs are derived from the inner cell mass (ICM) of the mammalian blastocyst, ESCs in standard culture conditions are transcriptionally heterogeneous containing dynamically interconverting subfractions primed for either of the two ICM lineages, Epiblast and PrE. Here, we find that differential regulation of cell cycle can tip the balance between these primed populations, such that naïve ESC culture promotes Epiblast-like expansion and PrE differentiation stimulates the selective survival and proliferation of PrE-primed cells. In endoderm differentiation, this change is accompanied by a counter-intuitive increase in G1 length, also observed *in vivo*. While fibroblast growth factor/extracellular signal-regulated kinase (FGF/ERK) signalling is a key regulator of ESC differentiation and PrE specification, we find it is not just responsible for ESCs heterogeneity, but also the inheritance of similar cell cycles between sisters and cousins. Taken together, our results indicate a tight relationship between transcriptional heterogeneity and cell cycle regulation in lineage specification, with primed cell populations providing a pool of flexible cell types that can be expanded in a lineage-specific fashion while allowing plasticity during early determination.

*For correspondence:
Joshua.brickman@sund.ku.dk

Present address: †Department of Pathology, Stanford University School of Medicine, Stanford, United States

## Editor's evaluation

This paper probes the link between lineage priming, lineage specification, and cell cycle in the ESCs. The authors report a number of interesting findings, including that: differential regulation of the cell cycle can tip the balance between populations of cells primed to different cell fate choices (PrE vs Epi), different culture conditions favor acceleration/stimulation of cell cycle of different cell populations, and that only a small population of cells from the original culture enters a differentiation process which is followed by selected expansion and/or survival of their progeny. They also observed that during endodermal specification (towards PrE), the cell cycle was shortened with a proportional relative increase of G1 phase length and that FGF activity is responsible for cell cycle synchronization and the inheritance of similar cell cycles between sisters and cousins. Together these finding indicate a close relationship between transcriptional heterogeneity and cell cycle regulation during lineage priming.

## Introduction

Naïve mouse embryonic stem cells (mESCs) are karyotypically normal, immortal cell lines derived from the inner cell mass (ICM) of pre-implantation embryos (*Martin, 1981*; *Evans and Kaufman, 1981*). ESCs are pluripotent, able to differentiate into all future cell lineages of the embryo proper, and able to retain pluripotency through successive rounds of self-renewing cell division (*Beddington and Robertson, 1989*; *Suemori et al., 1990*; *Lallemand and Brûlet, 1990*; *Martello and Smith, 2014*). In mouse, pluripotent stem cells can be derived from several stages of development and exhibit gene expression profiles matching these developmental stages (*Morgani et al., 2017*; *Riveiro and Brickman, 2020*).

Under specific conditions, mESCs can recapitulate several aspects of the ICM specification. When cultured in serum and the cytokine LIF, they constitute a dynamically heterogeneous cell culture model that contains populations that are primed, but not committed, for both primitive endoderm (PrE) and Epiblast. These populations exhibit biases in differentiation but will readily interconvert when left in self-renewing culture (*Canham et al., 2010*; *Morgani et al., 2013*; *Illingworth et al., 2016*). While these conditions involve serum, we recently defined a culture condition that supports heterogeneous ESC culture, and at the same time sustains effective propagation of pluripotency and high levels of germline transmission (*Anderson et al., 2017*). We refer to this culture condition as NACL for N2B27, Activin A, Chiron, and LIF. A similar defined media with the same components but an RPMI base (RACL) supports PrE differentiation and expansion (*Anderson et al., 2017*). As NACL ESC media is almost identical to the media used for PrE differentiation (RACL), this culture model is an ideal system for the comparison of self-renewal and differentiation.

Several time lapse studies have explored the role of heterogeneity in ESCs, and comparisons have been made between naïve culture in serum and media based on small molecule inhibitors of GSK3 and mitogen-activated protein kinase kinase (MEK), in addition to LIF (2i/LIF) (*Singer et al., 2014*; *Abranches et al., 2014*; *Filipczyk et al., 2015*; *Cannon et al., 2015*; *Hastreiter et al., 2018*). Taken together, these studies have focussed on the dynamic heterogeneity of pluripotency factors which are associated with supporting the pluripotent state and Epiblast specification *in vivo*. Although they have explored the role of pluripotency factors in lineage priming (*Strebinger et al., 2019*), they have not assessed the expression of lineage-specific markers, how cells progress into differentiation and the role of priming in differentiation. Here, we explore these questions with respect to PrE priming and differentiation.

*In vivo*, the segregation of PrE from Epiblast is regulated by the FGF/ERK pathway (*Chazaud et al., 2006*). The inhibition of this pathway results in an expansion of Epiblast identity and loss of PrE, both *in vivo* and *in vitro* (*Nichols et al., 2009*; *Yamanaka et al., 2010*; *Hamilton and Brickman, 2014*; *Saiz et al., 2016*). Based on the activity of this pathway in supporting PrE differentiation *in vivo*, inhibition of the kinase upstream ERK, MEK, with a pharmacological antagonist (PD03) is an important component of the defined ESC culture system known as 2i/LIF (*Ying et al., 2008*). Additionally, ERK activation is also associated with cell cycle progression and is thought to stimulate G1/S transition (*Yamamoto et al., 2006*; *Ter Huurne et al., 2017*). Although G1 lengthening has been functionally related to endoderm differentiation (*Calder et al., 2013*; *Pauklin and Vallier, 2013*; *Coronado et al., 2013*), it is not clear how ERK stimulation of cell cycle progression relates to its capacity to induce differentiation.

Here, we focus on the dynamics of PrE priming and differentiation exploiting both time lapse imaging and single-cell RNA-seq to link both events to regulation of the cell cycle. ESC culture (NACL) was found to support the more rapid proliferation of the Epiblast-primed population, and PrE differentiation (RACL) promotes the more rapid proliferation of cells primed for endoderm differentiation. We found that FGF signalling regulates cell states and proportions through both coordinating inheritance of cell cycle lengths as well as rates of endoderm priming. This functional regulation of cell cycle in heterogeneous cell cultures indicates that cell cycle synchronization supports self-renewal as well as preparing cells for further differentiation. Furthermore, we found that G1 length is adjusted independently from the cell cycle, pointing to a model where cells receive signals from cytokines during G1, while still actively proliferating. Taken together, our work suggests that cell cycle length is not only tightly regulated by the culture context, but that this coordination has a functional role in both heterogeneity and lineage choice.

## Results

### Single-cell RNA-seq of PrE differentiation reveals the presence of both endoderm and Epiblast-like populations in differentiation

To assess lineage-specific transcriptional heterogeneity in PrE differentiation and compare the events occurring *in vitro* to those occurring *in vivo*, we performed single-cell RNA sequencing by MARS-seq (*Jaitin et al., 2014*) on samples from 2i/LIF and days 1, 2, 3, 4, and 7 of *in vitro* PrE differentiation (*Figure 1A*). To track both endodermal and Epiblast lineages, we used a Sox2-GFP and Hhex-mCherry double fluorescent reporter cell line (*Figure 1A*; *Anderson et al., 2017*). Indexing based on reporter expression and plate-based MARS-seq enabled us to link individual transcriptomes to the cell types identified by Fluorescence-activated Cell Sorting (FACS) (*Figure 1—figure supplement 1A, B*). We collected equivalent numbers of different populations, that is we collected cells expressing different levels of SOX2 and *Hhex*, to ensure we had a good representation of the spectrum of cell types occurring in these *in vitro* cultures. As a result, the cell proportions based on cluster composition do not reflect the proportion of these cell types present at different time points during differentiation.

We detected nine clusters using unsupervised clustering (*Figure 1B*). Clusters 1, 5, and 6 were composed of 2i/LIF cells (*Figure 1—figure supplement 1C*, *Table 1*). Differentiating cells (from days 2 to 7) appeared to resolve into two branches: a PrE-like branch (clusters 3, 7, and 4) expressing progressively increasing levels of Hhex-mCherry (*Figure 1—figure supplement 1A*) and PrE markers such as *Dab2*, *Gata6*, *Pdgfra*, and *Sox17* (*Figure 1—figure supplement 2A*); and an Epiblast-like branch (clusters 0 and 2) with cells that appear to remain Epiblast-like, continuing to express pluripotency markers (Sox2-GFP, see *Figure 1—figure supplement 1B*; *Klf2*, *Nanog*, *Sox2*, and *Rex1*, see *Figure 1—figure supplement 2B*). As most Epiblast-like cells maintained expression of pluripotency markers (*Figure 1—figure supplement 2B*), but clearly have distinguished themselves from 2i/LIF ESCs, we refer to these cells as Non-Endodermal/Non-Differentiated (NEDiff) (*Figure 1B*). Cluster 8 was a tiny cluster that contained non-specific dying cells and thus it was discarded from downstream analysis.

We identified distinct endodermal signatures as early as day 2 (*Figure 1C*), suggesting that some cells at day 2 were already primed towards endoderm differentiation. The endodermal genes found at day 2 of the PrE branch (PrE in *Figure 1C*) were not upregulated at day 2 in the NEDiff branch (NEDiff in *Figure 1C*), supporting the notion that the transcriptional signature for PrE appeared on day 2 in a subpopulation of these cultures.

To compare these cell types to those induced *in vivo*, we compared our dataset with the already published scRNA-seq dataset of pre-implantation embryos (*Nowotschin et al., 2019*) using the Cluster Alignment Tool (CAT) (*Rothová et al., 2022*). Our day 7 PrE cluster 4 was most similar to PrE from E4.5, the earlier emerging PrE cluster 7, first appearing at day 2, resembled PrE from E3.5, while the rest of the clusters were most similar to E3.5 Epiblast (*Figure 1D*). Although cluster 3 was positioned at the beginning of the PrE branch, it aligned to Epiblast, continued to express Sox2 and showed low levels of Hhex. Based on these comparisons, we believe that our *in vitro* model is a good tool for deconstructing transcriptional signatures of differentiation and that an early PrE-like cell type arises by day 2 *in vitro*.

### PrE arises through early induction, followed by selective proliferation

We next sought to determine whether we could detect the emergence of these early (day 2) PrE-like cells and follow their differentiation using time lapse microscopy. We exploited a mESC line that couples a sensitive PrE reporter, Hhex-mCherry, with a second that enables lineage tracing, H2B-Venus (Hhex-3xFLAG-IRES-H2b-mCherry and pCAG-H2b-Venus, HFHCV) (*Illingworth et al., 2016*). These cells were used to follow mESC in defined PrE differentiation (RACL) (*Figure 2A*). To follow a significant number of cell cycles through differentiation, we acquired images every 20 min for 6 days (see *Video 1*).

As with the scRNA-seq time course, we started differentiation from a relatively uniform population of cells cultured in 2i/LIF and then differentiated them to PrE (see Methods, *Figure 2A*). When we analysed the fluorescence intensity distribution in this setup, we observed a distribution that suggested these cultures contained the same two populations as observed in the scRNA-seq: differentiated PrE and the NEDiff population that failed to progress towards endoderm. We clustered cell intensities using *k*-means clustering and used this clustering to assign identity in differentiation

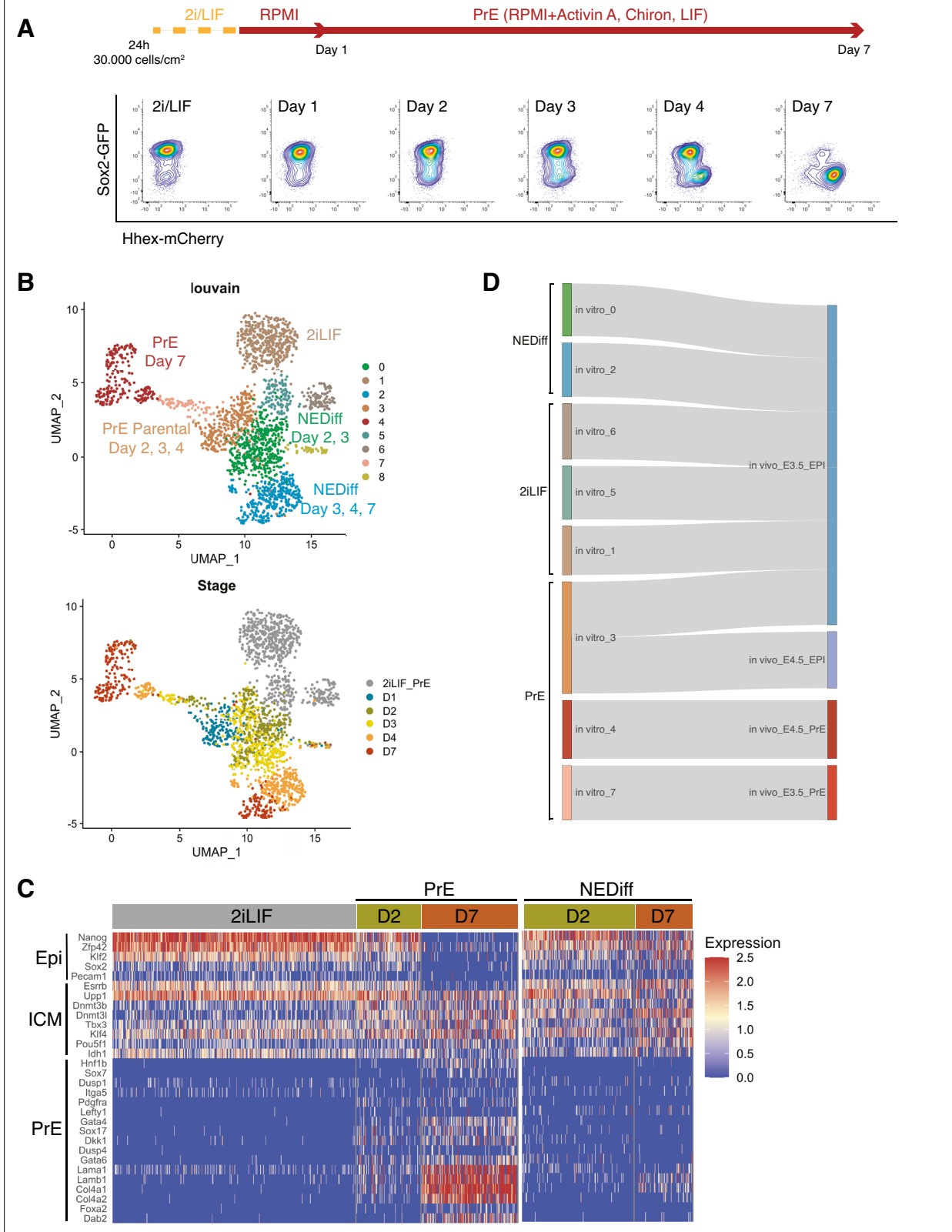

**Figure 1.** Transcriptome profiling of Primitive Endoderm (PrE) *in vitro* differentiation. (**A**) Schematic of the experiment. Cells were passaged twice in 2i/LIF and then plated in RPMI base media 24 hr before starting the experiment. Bottom panel: Flow cytometry plots showing the time points selected for single-cell RNA-seq. The fluorescent information of Sox2 and Hhex was recorded prior to sequencing. Cells from all the populations shown in the plots were collected for sequencing. (**B**) UMAP projection of the *in vitro* experiment showing nine identified clusters using Louvain (upper panel) and

*Figure 1 continued*

stages of differentiation (bottom panel). (**C**) Heatmap showing expression of selected Epi, Inner Cell Mass (ICM), and PrE markers in 2i/LIF, days 2 and 7 of differentiation. Left panel: PrE Diff branch. Right panel: NEDiff branch. Cells at day 2 in the PrE branch already are upregulating endoderm genes while the NEDiff cells are not. (**D**) Sankey plot visualizing cluster similarity comparison between identified *in vitro* clusters and *in vivo* (*Nowotschin et al., 2019*) experiment using the Cluster Alignment Tool (CAT).

The online version of this article includes the following figure supplement(s) for figure 1:

**Figure supplement 1.** Properties of cells collected for MARS-seq during Primitive Endoderm (PrE) *in vitro* differentiation.

**Figure supplement 2.** Lineage-specific markers expressed in single-cell RNA-seq clusters.

(*Figure 2B*, *Figure 2—figure supplement 1*). The identity and time evolution of PrE and NEDiff states were confirmed with GATA6 and NANOG staining (*Figure 2—figure supplement 2A, B*). Based on the lineage trees (*Figure 2C* and *Figure 2—figure supplement 1*) and increasing levels of PDGFRA expression (*Figure 2—figure supplement 2C*), the endodermal identity appeared around day 3 from a small founder population. This was also apparent when viewing individual time lapse movies (*Video 1*).

As possible contributors to the growing PrE population across *in vitro* differentiation, we aimed to distinguish between a purely selective or inductive process. Over the course of differentiation, selection would imply faster proliferation or enhanced survival of a founder population of primed PrE cells, while induction would be represented by endoderm cells converting continuously from the undifferentiated cell pool. To evaluate these possibilities, we assessed the behaviour of both endodermal and non-endodermal populations over time. We observed a decrease in the division time of the PrE cluster with cell division accelerating primarily on day 3 of differentiation (13.00 ± 4 hr for PrE and 20.00 ± 6 hr for NEDiff clusters on day 3) (*Figure 2D, E*, *Table 2*). This is the point in time when PrE clones became distinguishable from NEDiff cells in the time lapse video (*Video 1*) and suggested that culture conditions promoted endoderm specification by providing an environment that favoured the proliferation of a PrE-primed population. We also observed a higher rate of cell death and reduced survival (*Table 2*) in the NEDiff cells, suggesting an additional level of selective cell survival in the PrE progenitor population.

As these cultures started from a homogeneous population of 2i/LIF cells, we assumed some conversion to endoderm fate preceded the acceleration of proliferation and assessed the probability of cell state transition before and after day 3 (72 hr). We found that 12.6 ± 0.6% of the total population upregulated *Hhex* expression during the first 72 hr, thereby transitioning into the PrE cluster. However, after the 72-hr time point, only 1.9 ± 1.2% of the cells were able to transit between clusters. This suggested that competence to enter the endoderm lineage was lost after the first 3 days of the differentiation, indicating that proliferation was enhanced following lineage choice. Hence, even though only a small population of cells entered differentiation, the selective expansion of these cells allowed them to take over the final culture.

**Table 1.** Summary of the cell numbers in the different scRNA-seq clusters analysed. Clusters are annotated as NEDiff, PrE Parental, and PrE according to *Figure 1B*.

|  |  | 2i/LIF | Day 1 | Day 2 | Day 3 | Day 4 | Day 7 |
|---|---|---|---|---|---|---|---|
| 2i/LIF | Cluster 1 | 415 | 0 | 0 | 0 | 0 | 0 |
|  | Cluster 5 | 105 | 0 | 6 | 1 | 0 | 0 |
|  | Cluster 6 | 91 | 0 | 0 | 0 | 2 | 1 |
| NEDiff | Cluster 0 | 8 | 14 | 201 | 205 | 11 | 1 |
|  | Cluster 2 | 1 | 0 | 0 | 4 | 238 | 103 |
| PrE Parental | Cluster 3 | 2 | 139 | 78 | 76 | 3 | 2 |
| PrE | Cluster 7 | 0 | 7 | 32 | 29 | 4 | 0 |
|  | Cluster 4 | 0 | 0 | 1 | 2 | 44 | 162 |
|  | Total | 622 | 160 | 318 | 317 | 302 | 269 |

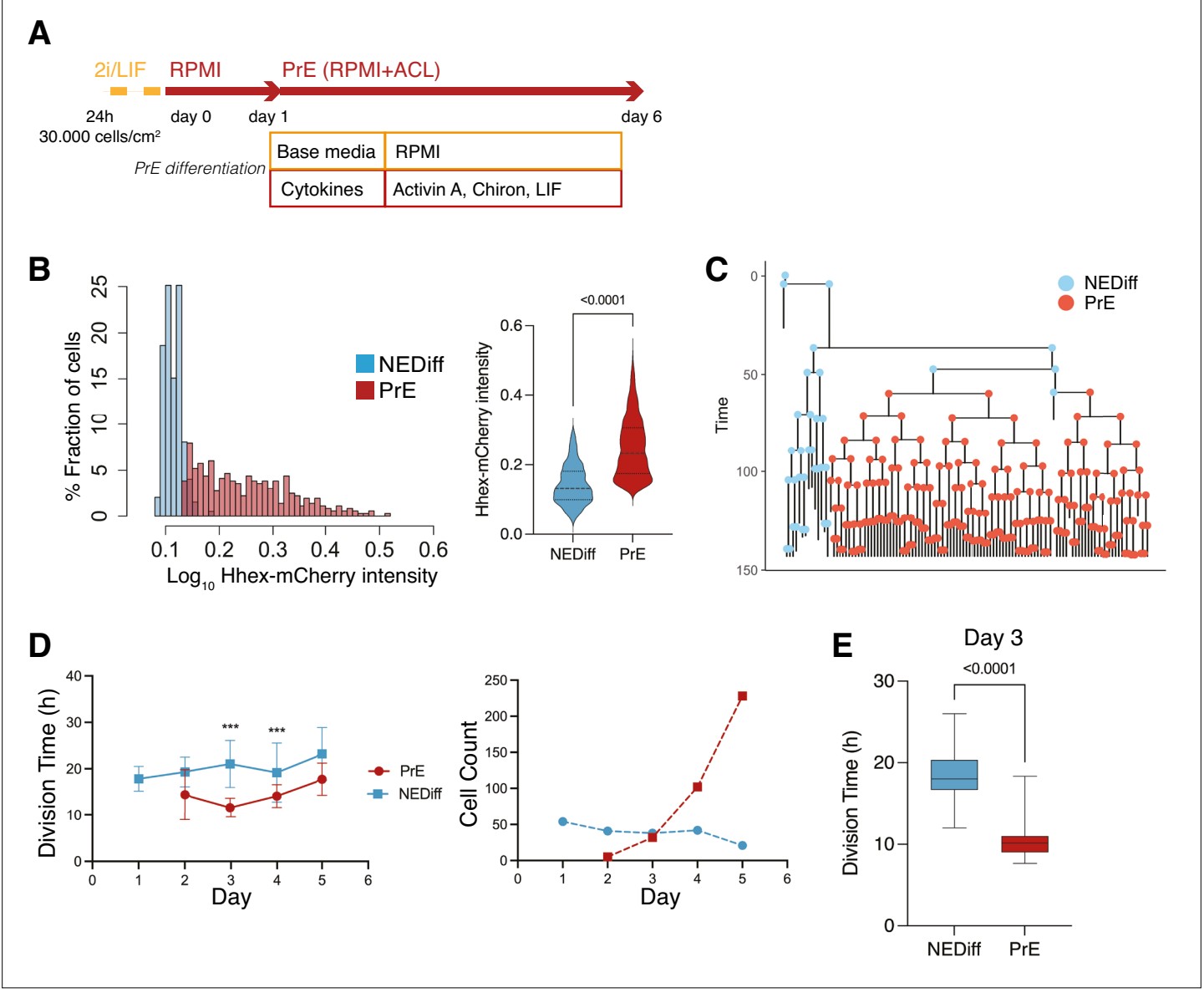

**Figure 2.** Time lapse of Primitive Endoderm (PrE) differentiation shows rapid proliferation of PrE-primed cells. (**A**) Schematic of the experimental setup. Cells were imaged for 6 days acquiring one time frame every 20 min. (**B**) The *Hhex* intensity distribution between the populations allowed us to separate PrE differentiated cells (PrE) from the Non-Endodermal/Non-Differentiated cells (NEDiff). p value <0.0001 Mann–Whitney test. (**C**) Example of a lineage tree showing how the PrE branch of the tree arises. The first and last generation were discarded from further analysis since the cell cycle information is not complete. All lineage trees collected in the PrE condition are shown in *Figure 2—figure supplement 1*; this example corresponds to Tree 10 in *Figure 2—figure supplement 1*. (**D**) Analysis of mouse embryonic stem cells (mESCs) division times and cell counts showed that cells that differentiate into PrE are dividing faster at the beginning of the differentiation process (day 3, left panel) and that selected survival likely takes place later in differentiation (days 4 and 5, right panel). ***p value <0.001. (**E**) Cell cycle length at day 3 shows a decrease in the PrE cells division time, compared to a slower dividing non-endodermal cluster (p value <0.0001 unpaired *t*-test).

The online version of this article includes the following figure supplement(s) for figure 2:

**Figure supplement 1.** Lineage trees in Primitive Endoderm (PrE) differentiation.

**Figure supplement 2.** Assessing Primitive Endoderm (PrE) differentiation *in vitro* and *in silico*.

To assess whether the combination of this small amount of lineage conversion followed by enhanced proliferation was sufficient to account for the behaviour of differentiating cultures, we sought to generate a simple model introducing the transition probability observed in time lapse (*t* in *Figure 3A*). This minimal set of parameters was not enough to recapitulate our lineage trees. We found that only

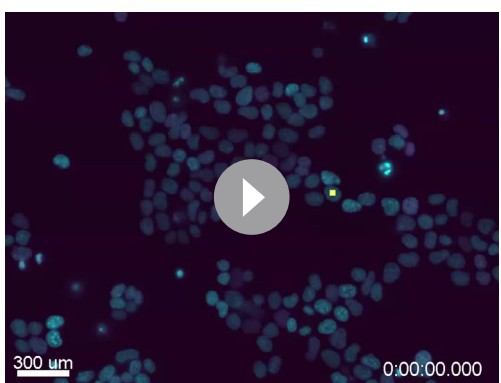

**Video 1.** Example of a tracked HFHCV colony during Primitive Endoderm (PrE) differentiation. Cyan is *H2B-Venus*, magenta is *Hhex-mCherry*. Yellow squares show the cell tracking. Scale bar is 300 μm.
https://elifesciences.org/articles/78967/figures#video1

after introducing both differential survival rates for PrE and NEDiff cells (*s* in *Figure 3A*), as well as different cell cycle contributions (*d* in *Figure 3A*) we could reproduce the cell populations observed. Thus, this minimal model predicted that selective cell death was also required to explain differentiation. To quantify the extent of this during differentiation, we determined the survival rates in our cultures. Before commitment took place at day 3, 86% of prospective PrE cells survived to generate colonies of PrE, whereas only 61% of the NEDiff survived (*Table 2*). This difference in survival was further enhanced following commitment at 72 hr and continuing throughout differentiation: from this point onwards, 92% of cells in the PrE cluster survived, whereas only 39% of the cells in the NEDiff cluster did (*Table 2*). We explored the possible contributions of proliferation and survival in our mathematical model by tuning the survival rates and the division times for

both populations in order to predict which parameter is more important in shaping the distributions of populations observed for each day (*Figure 3—figure supplement 1*). We found that on day 3, the cell cycle contribution has a more significant impact on the population composition, while cell survival has an increasing dominant impact on days 4 and 5. Therefore, it appeared that differentiation was selecting for committed endoderm at two levels: proliferation first, and cell survival later. Furthermore scRNA-seq suggested that the PrE population also accrued more endodermal identity as it expanded, such that endodermal gene expression not only became more homogeneous, but the level of endodermal transcription in individual cells increases from days 2 to 7 (*Figure 1C*), a process we refer to as progressive induction.

Lastly, given that we found an endodermal transcriptional signature in the scRNA-seq at day 2, we considered whether lineage priming and selection could be occurring prior to our assignment of differentiation (day 3). Thus, we took the dataset from day 2 of differentiation and asked whether low level reporter expression correlated with later differentiation by asking which cells would prospectively give rise to PrE (PrE Fate) and which would never become endoderm by day 3 (NEDiff Fate). We found that there was a tendency for PrE progenitor cells (PrE Fate) to express *Hhex* at higher levels (0.18) compared to cells that are non-endodermal progenitors (NEDiff Fate, 0.15) (*Figure 3B*, *Table 3*). To confirm the notion that these cells were functionally primed at day 2, we isolated *Hhex*-high and low populations from day 2 of the PrE differentiation by FACS and cultured them further

**Table 2.** Comparison of survival and proliferation during time lapse of Primitive Endoderm (PrE) differentiation.

Survival rate is calculated as a ratio between cells that survived and total cells. Death rate is the ratio between cells that died and total number of cells. As total cells, only cells with complete cell cycle information are considered. Division time (hours) is shown as median ± standard deviation.

| | PrE | | | | NEDiff | | | | |
|---|---|---|---|---|---|---|---|---|---|
| | Day 2 | Day 3 | Day 4 | Day 5 | Day 1 | Day 2 | Day 3 | Day 4 | Day 5 |
| Cells that survive | 5 | 32 | 102 | 228 | 54 | 41 | 38 | 42 | 21 |
| Cells that die | 1 | 4 | 7 | 24 | 20 | 42 | 16 | 40 | 57 |
| Cells total | 6 | 36 | 109 | 252 | 74 | 83 | 54 | 82 | 78 |
| Survival rate | 0.83 | 0.89 | 0.94 | 0.90 | 0.73 | 0.49 | 0.70 | 0.51 | 0.27 |
| Death rate | 0.17 | 0.11 | 0.06 | 0.10 | 0.27 | 0.51 | 0.30 | 0.49 | 0.73 |
| Division time (hr) | 13.3 ± 4 | 11.0 ± 3 | 13.0 ± 4 | 17.3 ± 4 | 16.3 ± 4 | 18.0 ± 7 | 20.0 ± 6 | 17.0 ± 7 | 22.0 ± 7 |

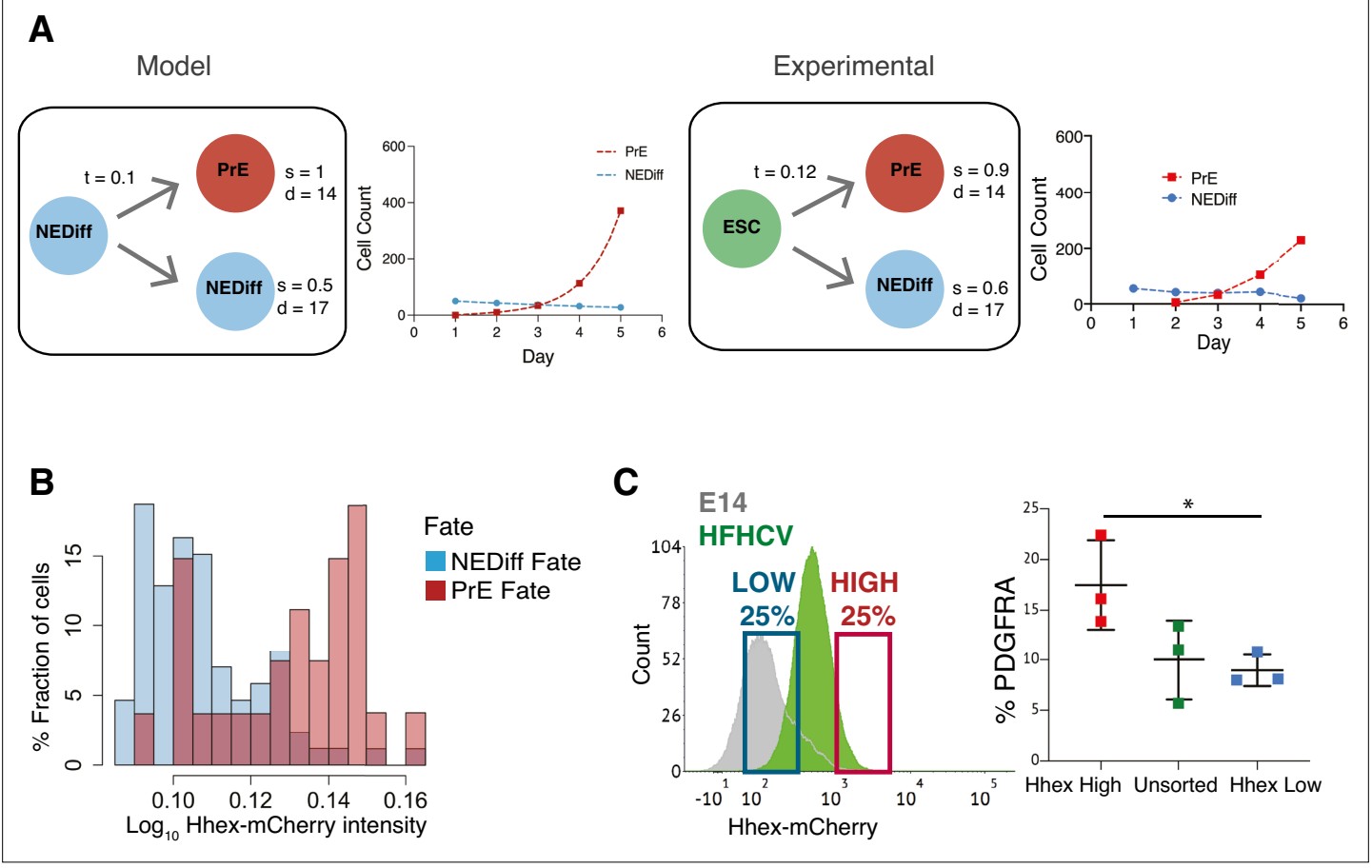

**Figure 3.** Analysis of Primitive Endoderm (PrE) progenitor cells demonstrates functional priming of Hhex-high cells. (**A**) A mathematical model that considers the difference in cell death rate between the two populations as well as the proliferation rate can recapitulate the PrE dataset collected (see Methods for description of the Mathematical Modelling). Based on the experimental dataset, we found the same survival rates predicted by the model. $t$ = transition rate, $s$ = survival rate, $d$ = division time (hr). (**B**) The NEDiff cluster at day 2 was separated into cells that will give rise to PrE (PrE Fate), and cells that eventually would not differentiate (NEDiff Fate). Analysis of the *Hhex* intensity distribution shows that cells that will give rise to PrE (PrE Fate) show higher *Hhex* intensity. Total cell number and fluorescence quantification shown in *Table 3*. (**C**) The High Hhex population from day 2 of PrE differentiation, isolated by FACS, shows improved PrE differentiation (scored as percentage of PDGFRA-positive cells), demonstrating the functional priming of these cells. *p value <0.05, Kruskal–Wallis test.

The online version of this article includes the following figure supplement(s) for figure 3:

**Figure supplement 1.** Algebraic model iterations.

in PrE differentiation conditions. Consistent with the time lapse analysis, we observed that the day 2 Hhex-high cells produced significantly more PDGFRA-positive PrE than either the Hhex-low or unsorted populations (*Figure 3C*).

**Table 3.** Analysis of Primitive Endoderm (PrE) parental cells.

Division time (hr) is shown as median ± standard deviation. Hhex-mCherry fluorescence (absolute units) is shown as median ± standard deviation.

|                   | PrE Fate    | NEDiff Fate |
| ----------------- | ----------- | ----------- |
| Cells             | 20          | 151         |
| Division time (hr) | 17.67 ± 6   | 17.33 ± 6   |
| Hhex-mCherry (a.u.) | 0.18 ± 0.03 | 0.15 ± 0.02 |

## PrE-primed mESCs in steady-state culture conditions

As the priming observed during *in vitro* PrE differentiation was similar to the spontaneous dynamic priming that occurs in standard ESC culture, but primed PrE remained a minority of the culture, we hypothesized that ESC culture might favour the proliferation of the undifferentiated or Epiblast-like population. As ESC culture can be supported by the same cytokines but with a different base media (*Anderson et al., 2017*), we reasoned the

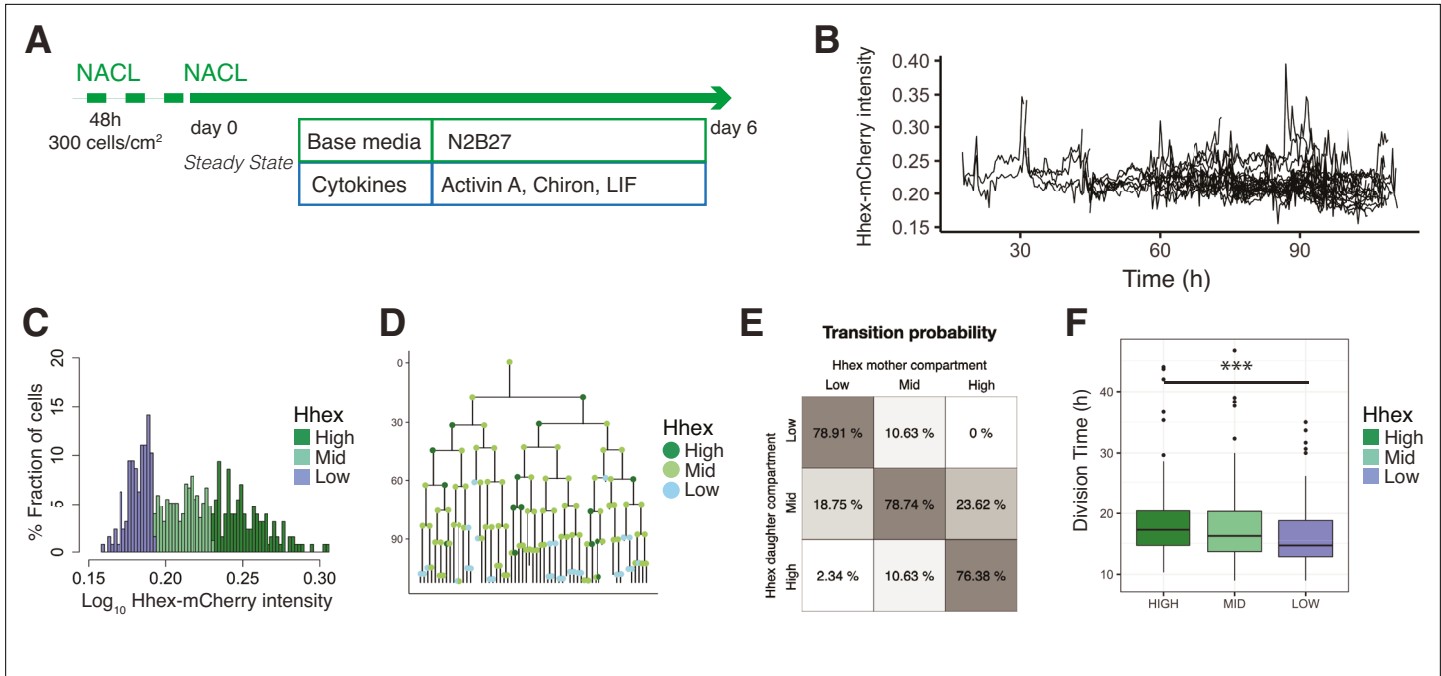

**Figure 4.** Single-cell quantification of *Hhex* expression in NACL uncovers a relationship between *Hhex* levels and cell cycle length. (**A**) Schematic of the experimental setup. Cells were plated 48 hr before starting the experiment. Cells were imaged for 6 days acquiring one time frame every 20 min. (**B**) Example of a cell trace (Time vs. Hhex-mCherry intensity) in the setup analysed. Cells survived and divided over 6 days without any apparent effect of cell death. Cells were entering and exiting higher and lower *Hhex* states without any apparent bias. (**C**) *Hhex* intensity distribution was divided into three compartments: High (includes cells above 75% percentile), Mid (between 25% and 75% percentiles), and Low (cells below 25% percentile). *Y*-Axis shows the percentage of cells that falls into each bin. See *Table 4* for total cell numbers per compartment. (**D**) Example of a lineage tree with the corresponding compartments of *Hhex*, by colour. The first and last generation were discarded from further analysis since the cell cycle information is not complete. All lineage trees collected in the NACL condition are shown in *Figure 4—figure supplement 2*. This example corresponds to Tree 1 in *Figure 4—figure supplement 2*. (**E**) Probability of cells to transition compartments between mother and daughter cells, quantified as percentage of cells over one generation. (**F**) The Low Hhex population divides significantly faster than the High Hhex. ***p value <0.001, Kruskal–Wallis test. The Mid Hhex population shows an in-between division time, suggesting a linear relationship between *Hhex* expression level and cell cycle length.

The online version of this article includes the following figure supplement(s) for figure 4:

**Figure supplement 1.** Heterogeneity in defined pluripotent stem cell culture.

**Figure supplement 2.** Lineage trees in pluripotent stem cell culture.

change in base media might underlie this difference. To explore this issue, we performed time lapse microscopy on the HFHCV mESC line in chemically defined NACL culture (*Anderson et al., 2017*; *Figure 4A*). We observed expansion of mESCs colonies from a small set of two to eight cells into several hundreds of cells, which produced imaging of up to seven generations (see *Video 2*, *Figure 4*).

We confirmed our previous observations (*Anderson et al., 2017*) that NACL culture supports a population of ESCs primed for PrE (Hhex-mCherry positive) and Epiblast (NANOG positive), within the OCT4-positive ESC population (*Figure 4—figure supplement 1A*). The distribution of *Hhex* expression that corresponded to these states was quite broad and showed a non-normal distribution (*Figure 4—figure supplement 1B*), similar to that observed for the constitutive lineage label H2B-Venus.

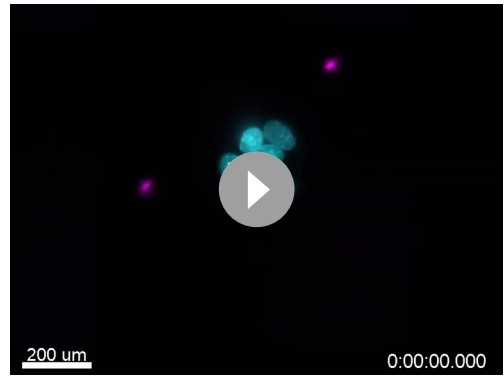

**Video 2.** Example of a tracked HFHCV colony in NACL. *H2B-Venus* is shown in cyan, magenta is *Hhex-mCherry*. Yellow squares show the cell tracking. Scale bar is 200 µm.
https://elifesciences.org/articles/78967/figures#video2

**Table 4.** Comparison of survival and proliferation between Hhex compartments.

Survival rate is calculated as a ratio of the number of cells that survived to the total number of cells. Death rate is the ratio of the number of cells that died to the total number of cells. As total cells, only cells with complete cell cycle information are considered. Division time (hr) is shown as median ± standard deviation. Residence times is shown as mean ± standard deviation of generations (or hours below) in which a cell stays in the same state.

| | NACL | | | PD03 | | |
|---|---|---|---|---|---|---|
| | High | Mid | Low | High | Mid | Low |
| Cells that survive | 127 | 254 | 128 | 48 | 191 | 252 |
| Cells that die | 12 | 35 | 34 | 14 | 55 | 55 |
| Cells total | 139 | 289 | 162 | 62 | 246 | 307 |
| Survival rate | 0.91 | 0.88 | 0.79 | 0.77 | 0.78 | 0.82 |
| Death rate | 0.09 | 0.12 | 0.21 | 0.23 | 0.22 | 0.18 |
| Division time (hr) | 17.3 ± 6 | 16.3 ± 6 | 14.7 ± 5 | 15.8 ± 5 | 14.0 ± 4 | 13.7 ± 4 |
| Residence time (generations) | 2.1 ± 1.27 | 2.2 ± 1.26 | 2.1 ± 1.34 | 1.6 ± 0.86 | 2.1 ± 1.22 | 2.9 ± 1.63 |
| Residence time (hr) | 37.9 ± 28.5 | 37.6 ± 28.9 | 29.6 ± 20.5 | 25.7 ± 18.5 | 29.1 ± 19.7 | 39.0 ± 23.9 |

Ideally, if the PrE- and Epiblast-like states were well defined, one would expect a bimodal distribution with one peak for each state. However, similar to previous reports for fluorescent reporters in ESC culture (*Abranches et al., 2014*; *Singer et al., 2014*), this was not observed. Based on previous analysis of NANOG expression (*Filipczyk et al., 2015*; *Hastreiter et al., 2018*), we divided the intensity distribution into compartments based on quartiles of the mean fluorescence intensity measurements. The compartments were called High (cells within the highest 25% intensities), Low (cells within the lowest 25% intensities), and Mid (the rest of the intensities) (*Figure 4C*). We constructed lineage trees using this compartmental distribution (*Figure 4D*, *Figure 4—figure supplement 2*).

The intersection between selection and induction during PrE differentiation time lapse described how a primed progenitor pool could give rise to a differentiated cell population. In NACL culture, primed cells are heterogeneously maintained, but the mechanisms that support and restrain this population are unknown. We propose that each cell can contribute to a population change by either dying, changing state, or proliferating. Therefore, we sought to determine how cell death, transitions, and cell cycle sustain naïve and primed populations in a dynamic equilibrium at steady state.

In NACL culture, we observed that 86% of cells survived, and cells divided every 17 hr on average. The level of cell death was not significantly different between the three compartments in our dataset (9% in the High, 12% in the Mid, and 21% in the Low, *Table 4*), so we went on to examine transitions. Based on the lineage trees (*Figure 4—figure supplement 2*), we estimated the probability that a cell would transit between these states by assuming that a cell undergoes transition if its mean intensity, averaged across cell cycle, changes to a different compartment than its mother cell. Altogether, we observed a similar behaviour between the Mid compartment and either the High or Low compartment, showing cells entering and exiting each state with the same probability (*Figure 4E*).

As this analysis describes the transition rate as the probability of transiting in each cell cycle, and therefore does not reflect the time required for a transition, we also calculated the transition rate based on the average time taken by a cell in one compartment to transition to another. We used the division time of each cell to express transitions per 24 hr unit and observed a similar rate (*Figure 4—figure supplement 1C*). In addition, we obtained residence times for each compartment to decipher whether a particular Hhex state was more stable than others, and we found no difference (*Table 4*). The residence time was calculated as the number of generations (or hours) that a cell remains in one state. To compare transitions in NACL to early differentiation, we measured the transition rates in RACL on days 1–3 of PrE differentiation and found that RACL promotes transitions towards higher Hhex compartments (*Figure 4—figure supplement 1D*).

Lastly, we investigated the influence of cell cycle. In RACL (PrE media), the Hhex-expressing primed population had a short cell cycle, but in NACL, where ESC self-renewal predominates, we found the opposite. Here, the Low Hhex cells exhibited significantly shorter division times than the High Hhex cells (14.67 ± 5 vs. 17.33 ± 6 hr, *Figure 4F*, *Table 4*). This result suggests a relationship between lineage priming and cell cycle length, with pluripotency being supported by media conditions that promote faster cycling of Epiblast-primed cells, while differentiation is supported by initially higher levels of PrE priming followed by cell cycle regulation and selection.

## MEK inhibition affects mESC priming by increasing the fraction of fast proliferating cells

While transition probabilities favour the expansion of the Mid compartment, shorter cell cycle times favour the Low compartment, and cell death appears indiscriminate. As FGF/ERK signalling both regulates PrE specification by directly altering enhancers and coordinates the cell cycle promoting G1/S transition (*Yamamoto et al., 2006*; *Hamilton and Brickman, 2014*) we asked whether inhibiting this pathway with PD03 acts to accelerate cell state transitions, cell cycle, or both. We quantified the changes in both *Hhex* expression and cell cycle in response to FGF/ERK inhibition by PD03 (*Figure 5A*, *Figure 5—figure supplement 1*), and the three compartments were defined using the same threshold intensities as for the NACL dataset.

Given the direct relation between ERK stimulation and *Hhex* upregulation (*Hamilton and Brickman, 2014*), we found unsurprisingly that cells in PD03 expressed significantly lower levels of *Hhex* than in the NACL control (*Figure 5B*). Thus, the proportion of cells in the Low Hhex compartment increased (*Figure 5C*). To understand if this increase stems from changes in cell death, proliferation, or switching probabilities, we compared these between the NACL and PD03 conditions. At the level of cell survival, PD03 treatment had little effect (80% cells that survived vs. 86% in NACL, *Table 4*). However, culture in PD03 produced significant alterations to both cell cycle length (*Figure 5D*) and cell state transitions (*Figure 5—figure supplement 2A*). While the influence of PD03 on cell state transitions is difficult to quantify robustly as PD03 can induce rapid cell state changes (*Hamilton et al., 2019*), we observed a high transition rate from the rare High Hhex cells found in PD03 conditions. We also observed shorter residence times for the High Hhex state in PD03, and longer times for the Low Hhex state (*Table 4*), suggesting that inclusion of PD03 in cell culture media suppresses the stability of the High state, such that any cell that escapes the Low Hhex and Mid compartments, rapidly returns.

We observed a shorter division time in the population of cells treated with PD03 (*Figure 5D* and *Table 4*, confirmed by bootstrap analysis, *Figure 5—figure supplement 2B*). Similar to the results observed in NACL, in PD03, cells in the Low Hhex compartment (Epiblast-like) had a shorter cell cycle than those in the other two compartments (*Figure 5E*, *Table 4*). However, when we compared the division time in the same compartment, but different media conditions (i.e. plus and minus PD03), we did not observe significant differences (*Figure 5E*). Together with an increased fraction of cells in the Low Hhex compartment (*Figure 5C*), this indicated that on a population level, PD03 decreased cell cycle length by increasing the fraction of fast proliferating cells in the Low Hhex (Epiblast-like) state. Thus, the change in cell cycle length observed in *Figure 5D* appears to be a population effect and not a result of PD03 regulating the cell cycle at the level of individual cells.

It has previously been shown that cell cycle lengths are inherited. This does not occur transgenerationally, as mothers and daughters do not have correlated cell cycle lengths, but sisters and cousins do (*Sandler et al., 2015*). Does this imply that synchronous cell cycle inheritance is a fundamental facet of lineage specification? In NACL, we found that cell cycle length was correlated between sisters and cousins, but not mother–daughter (*Figure 5—figure supplement 2C*). Given that PD03 produced a more homogeneous population of Low Hhex cells that divided faster, we hypothesized that cell cycle inheritance might therefore be better correlated. However, we found that cell cycle correlation between sisters and cousins was reduced or eliminated in the presence of PD03 (*Figure 5F*, *Figure 5—figure supplement 2D*). This capacity of PD03 to inhibit cell cycle correlations suggests that the mechanisms governing cell cycle entrainment are likely dependent on signalling downstream of MEK.

We considered two possible paradigms for ERK-dependent regulation of cell cycle synchronization: either the symmetric inheritance of a kinase activity, or the paracrine activity of a cytokine that drives MEK/ERK activation. FGF4 is a prominent cytokine known to activate the ERK pathway in ESCs

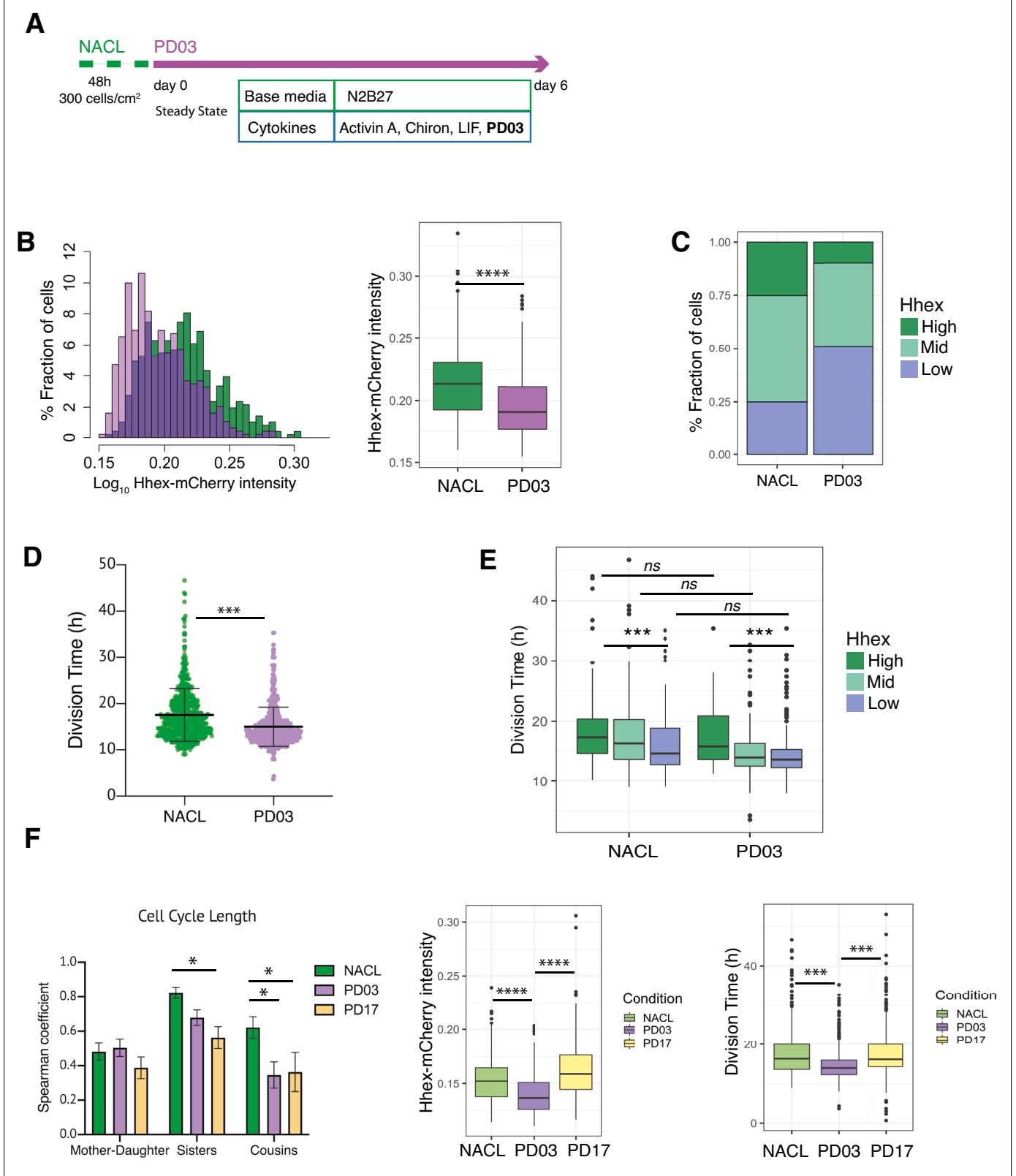

**Figure 5.** PD03 promotes expansion of the fast proliferating Low Hhex population. (**A**) Schematic of experimental setup. Cells were plated 48 hr before starting the experiment in NACL, and PD03 was added at the start of the time lapse. Cells were imaged for 6 days acquiring 1 time frame every 20 min. (**B**) *Hhex* intensity is significantly lower in the PD03 treated population. ****p value <0.0001, Wilcoxon test. (**C**) The lower intensity of *Hhex* is related to the higher fraction of cells in the Low Hhex population. The Low Hhex population increases from 25% to 50% when PD03 is added. (**D**)

*Figure 5 continued on next page*

*Figure 5 continued*

Mouse embryonic stem cells (mESCs) division time is significantly faster in PD03. ***p value <0.001, Mann–Whitney test. (**E**) Cell cycle in the Low Hhex compartment is faster in PD03 as well as in NACL. ***p value <0.001, Kruskal–Wallis test. (**F**) Left: Both PD03 and PD17 produce a loss in the cell cycle synchronization between sisters and cousins. All correlation plots are shown in *Figure 5—figure supplement 2*. *p value <0.05. Right: PD17 does not provide the same alterations in *Hhex* expression or division time that were generated by PD03. ***p value <0.001, ****p value <0.0001.

The online version of this article includes the following figure supplement(s) for figure 5:

**Figure supplement 1.** Lineage trees in pluripotent stem cell culture with PD03.

**Figure supplement 2.** Transitions and cell cycle synchronization in response to FGF/ERK inhibition.

and early embryos (*Kunath et al., 2007*; *Yamanaka et al., 2010*). Moreover, defined durations of FGF/ERK signalling can induce ESCs to adopt a PrE-primed state homogenously (*Hamilton and Brickman, 2014*). This suggests that the paracrine signalling with FGF and its receptor (FGFR) may be responsible for cell cycle synchronization in ESC culture. To test this idea, we followed ESC cell cycle and lineage heterogeneity in response to pharmacological inhibition of FGFR with PD17 (PD173074, an FGFR1/3 inhibitor). *Figure 5F* shows that culture in NACL with PD17 resulted in the same loss of cousin and sister correlation as that in response to inhibition of its downstream kinase MEK, further supporting the role for this pathway as a primary determinant of cell cycle synchronization. Furthermore, PD03 was both more effective at inducing Low Hhex populations and producing a general decrease in cell cycle length (*Figure 5F*, right; *Table 4*). However, despite the reduced effect of PD17 in ESC heterogeneity (*Figure 5F*, right), PD17 was at least as effective as PD03 in its capacity to block synchronization (*Figure 5F*, left; *Figure 5—figure supplement 2E*). Thus, even though PD17 containing cultures had significant proportions of Mid Hhex and High cells, that proliferated slowly, PD17 still blocked their synchronization, suggesting that this pivotal pathway independently regulates cell state transitions and cell cycle synchronization.

## G1 length regulation accompanies the changes observed in cell cycle

To further elucidate the differences in cell cycle length in this context, we performed time lapse using the FUCCI cell cycle reporter cell line (*Sakaue-Sawano et al., 2008*), engineered with an H2B lineage reporter (see Methods, *Video 3*). Using this approach, we measured division time, G1 length and G1 in relation to the total cell cycle length (G1 ratio) (*Figure 6*, *Table 5*). We measured these parameters in NACL, PD03, 2i/LIF, and PrE differentiation.

Comparing NACL, PD03, and 2i/LIF, PD03 did not seem to affect the average G1 length significantly, yet there appeared a small increase and clear outliers within PD03 and 2i/LIF cultured cells that showed an increased G1. Although the significance of this increased G1 length may require additional measurement, the increase in G1 length relative to cell cycle length was significant. Thus, in steady state, G1 either remains unaffected or is marginally increased despite the decreasing cell cycle time (*Figure 6A*, *Table 5*). We confirmed these unexpected changes by bootstrap analysis (*Figure 6— figure supplement 1*). Collectively our results suggest that G1 length is increased relative to the cell cycle in response to PD03. As we cannot combine the G1 reporter with *Hhex* lineage, we cannot assess whether it is the High Hhex or Low Hhex cells which present a longer G1. However, as the impact of PD03 on cell cycle is a consequence of stimulating the Low Hhex population, we presume an increase in this population accounts for the change in the ratio of G1 to the cell cycle. We therefore used division time to infer cell identity and found that fast dividing cells (Low Hhex) have a higher ratio of G1 to cell division than slow cells (0.15 ± 0.06 vs. 0.14 ± 0.04), suggesting Low Hhex cells have a longer G1. Taken together, our results suggest that PD03 promotes the faster proliferating Low Hhex population, but that the

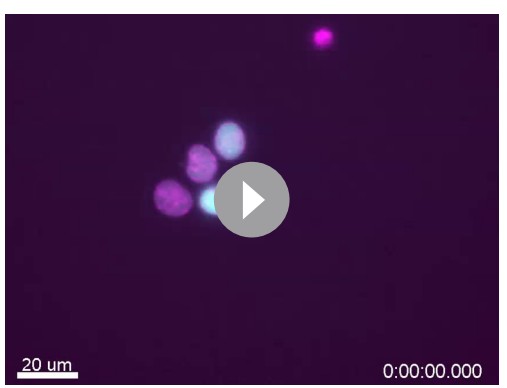

**Video 3.** Example of an imaged FUCCI colony in NACL. *mCherry-Cdt1* is shown in cyan, and *H2B-miRF670* is shown in magenta. Scale bar is 20 μm. https://elifesciences.org/articles/78967/figures#video3

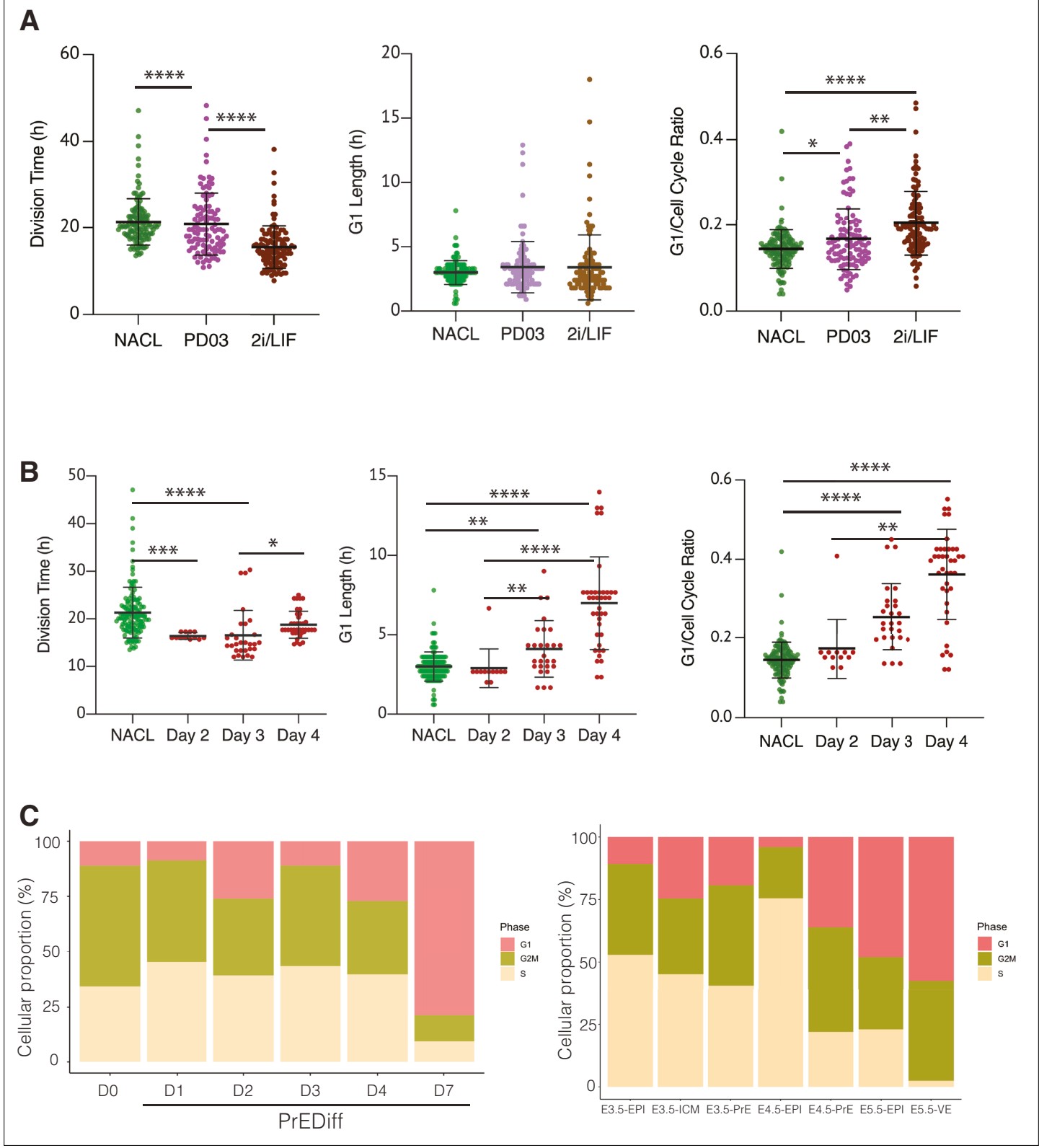

**Figure 6.** The regulation of cell cycle length during priming and differentiation involves G1 length. (**A**) PD03 addition significantly increases the ratio between G1 and division time (*p value <0.05, Kruskal–Wallis test), even though it does not significantly change the G1 length itself in FUCCI cells. Cells cultured in 2i/LIF for two passages show a faster cell cycle than when PD03 was added for a short period (**p value <0.01, ****p value <0.0001, Kruskal–Wallis test). (**B**) Primitive Endoderm (PrE) differentiated cells show a longer G1 phase and an increase in G1 ratio as they proceed along the

*Figure 6 continued on next page*

*Figure 6 continued*

differentiation process. Division time is faster at day 3 and then it slows down, consistent with the previous dataset. *p value <0.05, **p value <0.01, ***p value <0.001, ****p value <0.0001, Kruskal–Wallis test. (**C**) Proportion of cells that express G1, G2/M, or S signature transcriptional profiles. Left: *In vitro* dataset (this study). Right: *In vivo* dataset (*Nowotschin et al., 2019*).

The online version of this article includes the following figure supplement(s) for figure 6:

**Figure supplement 1.** Bootstrap analysis of the FUCCI dataset.

decreased cell cycle time in this population is not a result of stimulating G1/S transition, consistent with the role of this drug in inhibiting it (*Ter Huurne et al., 2017*).

We also assessed alternation in G1 during PrE differentiation (see *Video 4*). Since this cell line did not have a *Hhex* marker, we stained for GATA6 at the end of the time lapse and retrospectively identified PrE lineage trees, mapping the progenitor cells that would give rise to GATA6-positive colonies. Using this approach, we were able to locate the progenitors of fully differentiated PrE cells and determine their cell cycle parameters on different days of differentiation (*Figure 6B*, *Table 5*). Despite the decreased cell cycle length observed for endoderm committed cells at day 3, we observed a progressive increase in G1 itself and the G1 ratio in the PrE colonies (*Figure 6B*, right). This increase in G1 appeared in parallel with the increase in proliferative capacity that we assumed associated with endoderm commitment arising from days 3 to 4, further strengthening the apparently contradictory correlation between increased G1 and higher rates of cell division. While there is an enhanced ratio of G1 length in Epiblast-primed ESCs, the changes in G1 during differentiation are much more significant, with G1 itself doubling as cells progress in differentiation.

Finally, we assessed the proportion of cells that expressed G1 transcripts on our scRNA-seq dataset from *in vitro* differentiation. As shown in *Figure 6C*, we observed the same trend of cells entering G1 as they progress through the endoderm lineage. Although the time frame of our *in vitro* differentiation protocol is several days and *in vivo* development occurs more rapidly, we took advantage of the published early embryo dataset used to benchmark our differentiation (*Nowotschin et al., 2019*), and assessed whether a G1 trend also occurs *in vivo* during PrE specification. We found an increase in cells expressing G1 transcripts as they progressed into the endodermal lineage *in vivo*. Around the time of implantation E4.5, there was a robust difference in cell cycle phase between Epiblast, which was mostly in S phase, and PrE, where almost half of the cells were in G1. This suggests the coupling we observed between cell cycle and differentiation *in vitro* might be recapitulated in endoderm specification *in vivo*, although the shift in cell cycle would need to occur quite rapidly *in vivo* as the time scale for differentiation is much shorter.

## Discussion

In this paper, we describe a close relationship between cell cycle length, lineage priming, and differentiation. In ESC self-renewal *in vitro*, naïve Epiblast-like cells proliferate faster than those primed for PrE differentiation, but when the base media is changed, the tides are turned, and the PrE-primed population develops the proliferative advantage. This suggests that differential culture conditions

**Table 5.** Summary of dataset collected with the FUCCI reporter.

Division time (hours) is shown as median ± standard deviation. G1 length (hours) is shown as median ± standard deviation. G1 ratio is produced as the ratio between the G1 length and the total division time, and it is shown as median ± standard deviation.

| | | | | PrE | | |
|---|---|---|---|---|---|---|
| | NACL | PD03 | 2i/LIF | Day 2 | Day 3 | Day 4 |
| Division time (hr) | 20.4 ± 5 | 19.5 ± 7 | 15.0 ± 5 | 16.33 ± 1 | 15.0 ± 5 | 18.0 ± 3 |
| G1 length (hr) | 2.7 ± 2.5 | 3.1±1.9 | 3.0 ± 0.9 | 2.7 ± 1.2 | 4.3 ± 1.5 | 7.3 ± 2.9 |
| G1 ratio | 0.14 ± 0.1 | 0.15 ± 0.1 | 0.19 ± 0.1 | 0.16 ± 0.1 | 0.24 ± 0.1 | 0.40 ± 0.1 |
| Cells total | 126 | 104 | 110 | 12 | 29 | 39 |

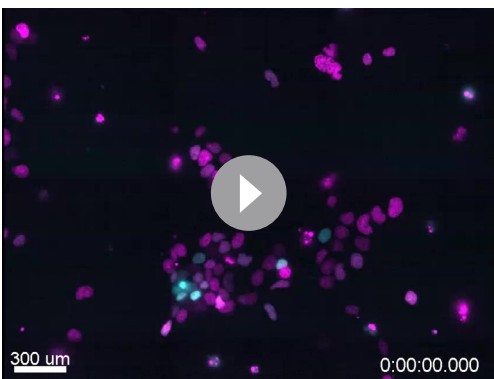

**Video 4.** Example of an imaged FUCCI colony during Primitive Endoderm (PrE) differentiation. *mCherry-Cdt1* is shown in cyan, and *H2B-miRF670* is shown in magenta. Scale bar is 300 µm.

https://elifesciences.org/articles/78967/figures#video4

identify the appropriate lineage-biased populations and stimulate their relative expansion. Our observation that similar alterations in cell cycle exist in peri-implantation development suggests that while this facet of differentiation is exploited *in vitro*, aspects of it could be a fundamental component of cell specification *in vivo*.

The differentiation process we define is both selective and progressively inductive. Heterogeneities arise, and these appear to be captured in early differentiation. However, the lineage-specific nature of these primed cells increases inductively (or progressively) such that endodermal identity accrues with each cell cycle. Primed cells that fail to enhance endodermal gene expression tend to die. By day 3 in differentiation, all the future founder cells in the culture are set, and further commitment does not occur. At this point, a combination of selective cell survival, increased rates of proliferation and enhanced endodermal transcription leads to the final differentiated cell populations. By the later time points in differentiation, the combination of single-cell sequencing, time lapse imaging, and mathematical modelling, suggests that cells that fail to enter the endoderm program selectively die, but at early time cell division dominates. Moreover, while differentiation promotes more rapid cell cycling, it also produces apparently contradictory increases in G1 length.

What are the factors that govern a cell's choice to differentiate? An increase in G1 would give cycling cells more time to respond to inductive signals, ultimately leading to increased responsiveness to differentiation cues. This would suggest that cells progress into early differentiation, lengthening their G1 phase, thereby improving accessibility to cues promoting commitment, and enabling signalling dependent transcription factors to accumulate ahead of replication. While the link between G1 phase and commitment of pluripotent stem cells into differentiation was first described in 1987 in embryonal carcinoma cells (*Mummery et al., 1987*), the recent development of the cell cycle reporter cell line FUCCI has allowed for more detailed findings in this field. It has been functionally demonstrated that the G1 fraction of ESCs has an enhanced capacity to respond to endoderm differentiation signals both in mouse (*Coronado et al., 2013*) and human (*Calder et al., 2013*; *Pauklin and Vallier, 2013*). In broad terms, these studies support the notion of G1 phase as a window where pluripotent stem cells can respond more effectively to differentiation cues, and specifically towards the endoderm lineage. Our findings that during the early stages of PrE differentiation cells lengthen their G1 phase as well as proliferate faster provide an explanation for how selected G1-responsive cells can also compete with other populations occupying the same niche. In self-renewing ESCs, PrE-primed Hhex-expressing cells are slowed. Culture conditions that promote a more homogeneous expansion of an Epiblast-like state include PD03, an inhibitor of G1/S transition, suggesting that the expansion of Epiblast-like cells could also involve similar cell cycle regulation. The notion that G1 phase lengthening accompanies differentiation has been shown in other cell lineages such as neural (*Lange et al., 2009*; *Roccio et al., 2013*), pancreatic (*Kim et al., 2015*; *Krentz et al., 2017*), and intestinal (*Carroll et al., 2018*).

In general, we described a tendency for increased rates of proliferation accompanied by increases in G1, and hence cells exhibiting a shorter cell cycle had a longer G1. While G1 progression has previously been linked to phosphorylation of Retinoblastoma protein (Rb) by ERK (*Ter Huurne et al., 2017*), this does not fit with our observations in differentiation or for PrE specification *in vivo*, where there is a progressive increase in G1 despite the robust activation of ERK signalling. Cell cycle progression is tightly regulated by the activity of cyclin-dependent kinases (CDKs), and inhibition of cyclin E-CDK2 produces G1 lengthening and spontaneous differentiation in pluripotent stem cells (*Filipczyk et al., 2007*; *Neganova et al., 2009*; *Coronado et al., 2013*). In mESCs, it has been shown that the ERK pathway is a regulator of G1 length upstream of CDK/cyclin complex which phosphorylates

Rb. Taking into account the heterogeneous distribution of cyclins and CDK proteins in the different cell types across development (*Wianny et al., 1998*), we propose that during PrE differentiation a different CDK/cyclin complex is regulating G1 phase length, and this complex is no longer a target of ERK phosphorylation in this context.

Mammalian cells share the striking quality of being synchronized in cell cycle length between members of the same generation (sisters and cousins), but not with the previous generation (mother–daughter). This was originally demonstrated in both a lymphoblast cell line and ESCs (*Sandler et al., 2015*; *Waisman et al., 2019*). Moreover, as alluded to above and suggested by previous work, this correlation might be determined based on factors that are inherited by related cells. This means that sister cells would inherit the same amount of determinant, which would be different from the previous generation (*Sandler et al., 2015*). Our observations suggest that this determinant is a factor located within the ERK pathway downstream of MEK, as culture with either the MEK inhibitor PD03 and/or the FGFR antagonist PD17 eliminates these correlations. Thus FGF/ERK regulation is likely a key nexus for this activity, and our observations suggest this activity is independent of its role in regulating differentiation. Whether this reflects a paracrine interaction of related cells with a common cytokine source or the level of receptor on the surface of these cells cannot be distinguished based on our observations. However, as ERK signalling is a known regulator of the G1/S transition, cell cycle synchronization could be mediated via this check point.

Given that cell cycle synchronization can be observed on both fast and slow proliferating cells, its relevance to differentiation is difficult to discern. If changes in proliferation are a response to differentiation cues, then synchronous inheritance could ensure that daughter cells retain an equivalent enhanced cell cycle state from their parents regardless of their position relative to the source of the signal. *In vivo*, where PrE differentiation occurs in a salt and pepper pattern in three dimensions, synchronization could enable cells to retain the memory of a signal regardless of their position and despite cell mixing. Moreover, if G1 represents the window of opportunity for cells to act upon signalling cues, we envision cell synchronization could represent inheritance of signalling opportunity, promoting similar durations of active transduction in clones of differentiating cells. If ERK is regulating both differentiation and cell cycle synchronization via G1, it is also possible that cell cycle synchronization is a consequence of ERK-mediated alterations to the temporal window over which a cell responds to cytokine signalling.

Differentiation *in vitro* occurs over several days, while in embryonic development the segregation of Epiblast from PrE occurs over a much shorter time scale, approximately 24 hr. While this difference limits the extent to which our observations *in vitro* can be extrapolated to embryonic development, the similarities between both transcriptome and G1 regulation are striking. Therefore, we speculate that cell cycle synchronization could facilitate a coordinated change in cell division time and G1 length in the context of embryonic lineage specification, enabling coordinated signalling responses.

Altogether, our data suggest that ESC cultures and their differentiation are complex models that involve an interplay of both priming-based selection and proliferation. We find that cell state changes are major drivers of culture-specific changes in proliferation that, in turn, contribute to the final state of a culture. Despite the increasing rates of proliferation, cells exploit G1 to integrate the signals coming from their environment and evaluate their choices. Although the time scale of the *in vivo* and *in vitro* differentiation is different, it appears that the use of increasing G1 to drive increasing levels of commitment is a fundamental facet of cell fate choice.

## Methods
### Mouse ESC culture
Mouse ESC lines were cultured in standard conditions in NACL as previously described in *Anderson et al., 2017*. For NACL + PD03 experiments, PD 0325901 (PD03) was added at the final concentration of 1 µM. For PrE experiments, cells were cultured previously in 2i/LIF containing CHI 99021 (Chiron), PD03, and LIF, and then in RACL as described in *Anderson et al., 2017*.

### Cell lines
Mouse ESCs lines used in this study were generated in E14JU ESCs. This line was derived from a 129/Ola background.

The cell line HFHCV was previously reported in *Illingworth et al., 2016*. It contains Hhex-3xFLAG-IRES-H2b-mCherry and a pCAG-H2b-Venus vector (HFHCV). Using this cell line *Hhex* transcription can be visualized by translational amplification of *Hhex*, which drives the expression of the monomeric fluorescent protein mCherry (*Canham et al., 2010*; *Morgani et al., 2013*; *Illingworth et al., 2016*).

The FUCCI-H2B-miRF670 reporter construct was constructed as follows. The H2B and miRFP670 sequences were fused by overlap extension PCR. The miRFP670 sequence was cloned from pY42-pmiRFP670AAA-NLS-Myc, a kind gift from YH Kim. The resulting fragment H2B-miRFP670 was inserted into the PCR-Blunt II-TOPO backbone by TOPO cloning. The H2B-miRFP670 fragment was digested with the *CpoI* (*RsrII*) and *KlfI* enzymes and inserted upstream of the Hygromycin resistance gene cassette of the ES-FUCCI plasmid (Addgene #62451). FCXC^T2 ESCs (*Hamilton and Brickman, 2014*) were electroporated with *BglII*-linearized FUCCI-H2B-miRFP670 DNA (25 μg), and stable transfectants were selected for with hygromycin (125 μg/ml). Cell lines were validated by flow cytometry, fluorescence microscopy, and karyotyping.

All cell lines have been routinely tested for mycoplasma.

## Time lapse

Mouse ESC lines were cultured in NACL or 2i/LIF medium on Laminin 511 (BioLamina)-coated 8-well slides (Ibidi) and imaged at 20 min intervals for 6 days in mCherry and Venus fluorescent light channels, in 5% $CO_2$ and 20% $O_2$ at 37°C under a Deltavision Widefield Screening microscope. ESCs were seeded at 300 cells/cm$^2$ 48 hr before the beginning of the time lapse for NACL and PD03 experiments, or at 30,000 cells/cm$^2$ 24 hr before for PrE differentiation. For PD03 experiments, cells were cultured in NACL for at least two passages and PD03 was added just before starting the time lapse. For PrE experiments, cells were cultured in 2i/LIF for 2 passages and then changed to RPMI minimal medium just before starting the experiment. Activin A, Chiron, and LIF (RACL) were added 24 hr after the time lapse started. In all experiments, the media was changed every day of the time lapse. To test that the laminin coating was not affecting the behaviour of the HFHCV cell line, we sorted Hhex-high and low populations seeded in both gelatine and laminin and analysed them 24 and 48 hr later to test that their re-equilibration rates are the same.

## Flow cytometry

Cells were collected by trypsinization and stained against a marker of undifferentiated mESCs, Pecam-1 (BD Biosciences, APC-conjugated, 551262; 1:200), or a marker for PrE differentiated cells, PDGFRA (BD Biosciences, APC-conjugated, 562777; 1:200), and DAPI (Molecular Probes, D1306, 1 μg/ml) to exclude dead cells. mESCs were stained for 15 min at 4°C before being washed and resuspended in FACS buffer (10% Fetal Calf Serum (FCS) in phosphate-buffered saline) with DAPI. Flow cytometry analysis was carried out using a BD LSR Fortessa, and flow cytometry sorting was carried out in a BD FACS Aria III. Data analysis was carried out using FCS Express 6 Flow software (De Novo Software) by gating on forward and side scatter to identify a cell population and eliminate debris, then gating DAPI negative, viable cells before assessing the levels of GFP, mCherry, or APC.

## mESCs immunostaining

Mouse ESCs were cultured in 8-well slides (Ibidi). ESC immunostaining was carried out as previously described in *Canham et al., 2010*. The following antibodies were used: anti-NANOG (eBioscience, 14-5761, 1:200), anti-OCT4 (Santa Cruz, sc-5279, 1:200), and anti-GATA6 (Cell Signalling Technologies, 5851, 1:1600). Secondary antibodies used are from the Alexa Fluor series (Molecular Probes, Thermo Fisher). mESCs were imaged using a Deltavision Widefield Screening microscope.

## Cell tracking

We performed manual cell tracking using Imaris v9.5 (Bitplane). Nuclei were segmented using the H2B marker, and we measured the Hhex-mCherry fluorescence intensity of a circular area of 50 μm diameter inside the segmented nuclei. For each area measured, we took the median fluorescence intensity as the measure for that given data point. Intensity measurements were linked to its time point and lineage, allowing us to infer the division time for each cell that was tracked, as well as the expression level of *Hhex* in each time point. Only cells with completed cell cycle information were used for cell cycle and compartment analysis.

## Data analysis

Data mining was performed with Matlab. A script was generated to take the separated output data frames from Imaris and convert into a single file containing all the dataset information, by cell. This file also contains the lineage tree information, mean fluorescence intensities, and division times. Alive and dead cells, as well as cells without complete cell cycle information, are located into groups for further analysis. Another file organized by time frame is generated in order to perform time dynamics analysis.

Statistical analysis and plotting were performed in R. *Hhex* compartments were determined by dividing the Hhex-mCherry intensity distribution into quartiles. All cells above the first quartile (0.1648) are considered High, and all cells below the third quartile (0.1376) are Low. The same quartiles are used for the PD03 dataset (*Table 4*). The PrE dataset was divided into PrE cluster and NEDiff cluster using *k*-means clustering.

Plots were created with R ggplot2 package, and GraphPad Prism.

## Mathematical modelling in Figure 3

The model describes the growth of the NEDiff population and PrE populations as exponential growth functions. Starting from $N$ NEDiff cells and 0 PrE cells, NEDiff cells transition to PrE at a constant rate $t$. Each generation number is described as $24\left(x-1\right)/d_1$ where $d_1$ is the division time of the NEDiff cells (in hours) and $x$ is the number of days, starting with 1. The number of NEDiff cells can then be described as follows, where $s_1$ is their survival rate:

$$N_{NEDiff} = N\,\left[2s_1\left(1-t\right)\right]^{\frac{24(x-1)}{d_1}}$$

PrE cells are described as the sum of cells that transition from NEDiff to PrE at each generation with rate $t$ and consequently divide with division time $d_2$ and survive with survival rate $s_2$:

$$N_{PrE} = \sum_{i=1}^{g} N\left(2s_1\right)^i\left(1-t\right)^{i-1} t\left(2s_2\right)^{\frac{(g-i)d_1}{d_2}} = 2\,N\,t\,s_1\,\frac{(2s_2)^{\frac{d_1}{d_2}\frac{24(x-1)}{d_1}} - \left(2s_1\left(1-t\right)\right)^{\frac{24(x-1)}{d_1}}}{2s_1\left(t-1\right)+\left(2s_2\right)^{\frac{d_1}{d_2}}}$$

The sum was solved using Wolfram Alpha Mathematica.

## scRNA-seq analysis

Single-cell libraries were sequenced using Illumina NextSeq 500. Pre-processing steps and quality control were done as described previously in *Jaitin et al., 2014*; *Rothová et al., 2022*. In short, the reads were aligned using HISAT (version 0.1.6) and mapped to mouse mm9 genome. Further downstream analysis was done using scanpy (version 1.4.6). After filtering 2028 cells and 17,769 genes were normalized and log transformed. We identified 2000 highly variable genes using seurat flavour. Genes were further scaled to mean variance followed by Principal Component Analysis (PCA) and Uniform Manifold Approximation and Projection (UMAP) dimension reduction. Neither cell cycle regression nor batch correction were necessary. Finally, we used unsupervised Louvain clustering with resolution set to 0.8 and which identified nine overall clusters (04_analysis.Rmd). Cell cycle was estimated using Seurat's (4.0.1) *CellCycleScoring* function.

## Cluster Alignment Tool

To estimate similarity between clusters we used CAT (*Rothová et al., 2022*). First, the tool normalized the datasets using non-zero median which removes the influence of outlier genes. Both datasets are subset for unique and common genes. Next, Euclidean distance is calculated between each pair of clusters by randomly sampling cells with replacement to cluster size. This step is repeated 1000 times generating two distributions. If the distance is significant (sigma = 1.6 representing *p* value = 0.05) we define these clusters transcriptionally similar.

## Code availability

We deposited the original MATLAB scripts behind the data mining and mathematical models on Github: https://github.com/SilasBoyeNissen/TranscriptionalHeterogeneityAndCellCycleRegulationAs

CentralDeterminantsOfPrimitiveEndodermPriming, (copy archived at swh:1:rev:4ccfdb66c16705ac-4171b284235c7d90c589b668, *Nissen, 2022*).

All scRNA-seq analyses were also uploaded to Github at http://github.com/brickmanlab/perera_et_al_2022/, (copy archived at swh:1:rev:b1adac3f247d4a106e5fb6c0cb880a2402dc38d6, *Proks, 2022*).

## Data availability

The scRNA-seq data used in this study have been deposited in the Gene Expression Omnibus, https://www.ncbi.nlm.nih.gov/geo/, the accession number is GSE200534. Previously published *Nowotschin et al., 2019* data that were used here are available under accession number GSE123046.

## Materials availability

All materials will be available upon reasonable request to the corresponding author.

## Acknowledgements

We thank James Briscoe for critical discussion, Jose Alejandro Herrera Romero for bioinformatics advice, YH Kim for the RFP670 plasmid, and the Brickman laboratory members for critical discussion and reading of this manuscript. We thank Helen Neil, Magali Michaut, and the reNEW Genomics Platform for technical expertise, support, and the use of instruments. We thank Jutta Bulkescher and the reNEW Imaging Platform for training, technical expertise, support, and the use of microscopes. We thank Gelo dela Cruz, Paul van Dieken, and the reNEW Flow Cytometry Platform for technical expertise, support, and the use of instruments. This work was supported by grants from the Lundbeck Foundation (R198-2015-412); Independent Research Fund Denmark (DFF-8020-00100B); the Danish National Research Foundation (DNRF116); MP was supported by a PhD studentship from the Lundbeck Foundation (R286-2018-1534); RSM was supported by a fellowship from the Lundbeck Foundation (R303-2018-2939); The Novo Nordisk Foundation Center for Stem Cell Medicine (reNEW) is supported by a Novo Nordisk Foundation grant number NNF21CC0073729, and previously NNF17CC0027852.

## Additional information

### Competing interests

Joshua M Brickman: Reviewing editor, *eLife*. The other authors declare that no competing interests exist.

### Funding

| Funder | Grant reference number | Author |
| --- | --- | --- |
| Lundbeckfonden | R198-2015-412 | Joshua M Brickman |
| Danish Agency for Science and Higher Education | DFF-8020-00100B | Joshua M Brickman |
| Danish National Research Foundation | DNRF116 | Joshua M Brickman Ala Trusina |
| Lundbeckfonden | R286-2018-1534 | Marta Perera |
| Lundbeckfonden | R303-2018-2939 | Rita S Monteiro |
| Novo Nordisk Fonden | NNF21CC0073729 | Joshua M Brickman |
| Novo Nordisk Fonden | NNF17CC002785 | Joshua M Brickman |

The funders had no role in study design, data collection, and interpretation, or the decision to submit the work for publication.

## Author contributions
Marta Perera, Conceptualization, Data curation, Formal analysis, Investigation, Methodology, Writing - original draft, Writing – review and editing; Silas Boye Nissen, Data curation, Software, Investigation, Writing – review and editing; Martin Proks, Data curation, Software, Formal analysis, Writing – review and editing; Sara Pozzi, Formal analysis, Investigation; Rita S Monteiro, Formal analysis, Methodology; Ala Trusina, Conceptualization, Supervision, Funding acquisition, Writing – review and editing; Joshua M Brickman, Conceptualization, Supervision, Funding acquisition, Writing - original draft, Writing – review and editing

## Author ORCIDs
Marta Perera http://orcid.org/0000-0002-9512-1109
Silas Boye Nissen http://orcid.org/0000-0002-9473-4755
Ala Trusina http://orcid.org/0000-0003-1945-454X
Joshua M Brickman http://orcid.org/0000-0003-1580-7491

## Decision letter and Author response
Decision letter https://doi.org/10.7554/eLife.78967.sa1
Author response https://doi.org/10.7554/eLife.78967.sa2

# Additional files

## Supplementary files
• MDAR checklist

## Data availability
The scRNA-seq data used in this study have been deposited in the Gene Expression Omnibus and are available under the accession number GSE200534. Previously published Nowotschin et al., 2019 data that were used here are available under accession number GSE123046.

The following dataset was generated:

| Author(s) | Year | Dataset title | Dataset URL | Database and Identifier |
| --- | --- | --- | --- | --- |
| Perera M | 2022 | Transcriptional Heterogeneity and Cell Cycle Regulation as Central Determinants of Primitive Endoderm Priming | https://www.ncbi.nlm.nih.gov/geo/GSE200534 | NCBI Gene Expression Omnibus, GSE200534 |

The following previously published dataset was used:

| Author(s) | Year | Dataset title | Dataset URL | Database and Identifier |
| --- | --- | --- | --- | --- |
| Nowotschin, et al | 2019 | The Emergent Landscape of the Mouse Gut Endoderm at Single-Cell Resolution | https://www.ncbi.nlm.nih.gov/geo/GSE123046 | NCBI Gene Expression Omnibus, GSE123046 |

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
