## [Editor Report]

This paper probes the link between lineage priming, lineage specification, and cell cycle in the ESCs. The authors report a number of interesting findings, including that: differential regulation of the cell cycle can tip the balance between populations of cells primed to different cell fate choices (PrE vs Epi), different culture conditions favor acceleration/stimulation of cell cycle of different cell populations, and that only a small population of cells from the original culture enters a differentiation process which is followed by selected expansion and/or survival of their progeny. They also observed that during endodermal specification (towards PrE), the cell cycle was shortened with a proportional relative increase of G1 phase length and that FGF activity is responsible for cell cycle synchronization and the inheritance of similar cell cycles between sisters and cousins. Together these finding indicate a close relationship between transcriptional heterogeneity and cell cycle regulation during lineage priming.

---

## [Decision Letter]

**Decision letter after peer review:**

Thank you for submitting your article "Transcriptional Heterogeneity and Cell Cycle Regulation as Central Determinants of Primitive Endoderm Priming." for consideration by *eLife*. Your article has been reviewed by 3 peer reviewers, and the evaluation has been overseen by a Reviewing Editor and Marianne Bronner as the Senior Editor. The reviewers have opted to remain anonymous.

Essential revisions:

This paper probes the link between lineage priming, lineage specification, and cell cycle in the ESCs. The authors report a number of interesting findings, including that: differential regulation of the cell cycle can tip the balance between populations of cells primed to different cell fate choices ( PrE vs Epi), different culture conditions favor acceleration/stimulation of cell cycle of different cell populations, and that only a small population of cells from the original culture enters a differentiation process which is followed by selected expansion and/or survival of their progeny. They also observed that during endodermal specification (towards PrE), the cell cycle was shortened with a proportional relative increase of G1 phase length and that FGF activity is responsible for cell cycle synchronization and the inheritance of similar cell cycles between sisters and cousins. The reviewers agree that this work is interesting and well done and would be appropriate for publication in *eLife* once the below concerns have been addressed.

1) There was concern about the massive amount of cell death in the time-lapse data underlying Figures 2 and 3. The media-specific effects on cell survival need to be explained and discussed in a revised manuscript.

2) There was also concern about the extent to which parallels can be drawn between findings in ESCs and embryonic development. For example, ESCs have a normal growth phase and so all these cells in culture have the same size but early embryo cleavages occur without a growth phase. Similarly, the timeline of PrE vs Epi specification in vivo and in vitro are completely different. In embryos, PrE is specified within 24h, whereas in vitro it takes 6 days. Thus it is difficult to see how these two timelines together with the different cell cycle lengths can be reliably compared. These limitations should be discussed in a revised manuscript.

3) At least for the PrE differentiation conditions, the authors need to better emphasize the strong survival effects of the media and discuss the relative contributions of selective cell death and cell cycle changes for generating a homogeneously differentiated PrE culture.

4) The authors should address whether RACL is purely selective through changes to cell survival and cell cycle, or does it in addition trigger an increase in priming towards the Hhex-positive population before the acceleration of the cell cycle between days 2 and 3? It would be informative to compare transition rates towards the Hhex-high state between NACL and RACL to address this question

5) The authors should clarify their reasoning as to why cell cycle synchronization is relevant for the differentiation of a culture, or lineage priming (Lines 72, 73, 235). How does the absence or presence of cell cycle synchronization between daughter cells impact the composition of cell populations at a steady state or upon differentiation, as both long and short cell cycles appear to be equally well correlated?

6) The use of the term "differentiation" is somewhat confusing (,e.g. in line 136 "differentiation proceeds by providing an environment that favored the proliferation of a PrE-primed population"). As this is an initially heterogeneous cell population transiting to a more homogeneous, different state there may be a better way of describing this without terming it differentiation.

7) The data underlying the table in Figure S5C seem important as they address how state transitions shape the population structure in addition to the cell cycle properties of the different compartments. This should be discussed more extensively in a revised manuscript and perhaps moved to the main Results section. It would also be interesting to quantify residence times and rates of state transitions in units of time-1. These analyses could then be quantitatively compared to corresponding transitioning rates in the presence of PD03 (current Figure S8A, and lines 221 – 225)

8) In Line 24 of the abstract the authors state that there is a lengthening of G1 in PrE-favouring conditions, but in the main body of the text it is the proportion between G1 and the rest of the cycle that is changing, ( "Thus, G1 either remains unaffected or is marginally increased despite the decreasing cell cycle time (Figure 6A, Table S5)"). Similar claims about the lengthening of the G1 are made in the Discussion. The authors should ensure that the result is described consistently throughout the manuscript

9) In Line 123 of the Results section, the authors state that: "The identity and distribution of PrE and NEDiff states were confirmed with GATA6 and NANOG staining (Figure S4A)." but the data presented by the authors are not sufficient to support this claim. Pairwise correlation between individual marker intensity needs to be calculated to prove this point with appropriate controls (for example with the use of the same primary antibody with two different secondary antibodies)

10) In Line 274 of the Results section, the authors state: "As we cannot combine the G1 reporter with Hhex lineage, we cannot assess whether it is the High Hhex or Low Hhex cells which present a longer G1." – I wonder why the authors cannot use here the same combination of the time-lapse plus immunostaining images as in the earlier experiments.

11) The figure legend for figure S1 needs more detail so that it can be followed without cross-referencing with the main text and figure 1.

---

## [Author Response]

Essential revisions:1) There was concern about the massive amount of cell death in the time-lapse data underlying Figures 2 and 3. The media-specific effects on cell survival need to be explained and discussed in a revised manuscript.

We acknowledge that there is a large amount of cell death in the beginning of differentiation and believe this could be a response to changes in media and cytokines. This is also observed in other differentiation and reprogramming protocols (Ying and Smith 2003; Hayashi et al., 2011; Yasunaga et al., 2005; Argelaguet et al., 2019; Rugg-Gunn 2022).

While our original analyses were focused on cell cycle, we did not mean to imply that survival is irrelevant to selecting the correct populations at the end of differentiation. Although there is clearly an increase in NEDiff cell death, these rates fluctuate during differentiation (Table 2) and it is difficult to make any hard conclusions about the timing of selection. In addition, we find no signature for apoptosis in all scRNA-seq clusters associated to early NEDiff. However, we thank the reviewers for raising this issue and we address the issue of survival more extensively in the revised manuscript (lines140-142, and 342-347, in addition to a new modelling section described below).

To further explore when relative survival is most important for effective differentiation, we have reformulated our model. Our new modelling results show that changes in the survival rates have stronger impact shifting the cell proportions at later stages of differentiation. Phase diagrams in Figure 3- Supplement 1 show that at day 5 reaching an appropriate proportion of PrE requires less change in the survival rates than in doubling times, suggesting that survival can have greater impact on the ratio than cell division. However, at day 3, the modelling suggests the opposite, consistent with a first wave of a cell cycle regulation and PrE fastest division time (Figure 2D). We have discussed this in the Results section, lines 162-168.

We have added survival alongside proliferation to the abstract.

2) There was also concern about the extent to which parallels can be drawn between findings in ESCs and embryonic development. For example, ESCs have a normal growth phase and so all these cells in culture have the same size but early embryo cleavages occur without a growth phase. Similarly, the timeline of PrE vs Epi specification in vivo and in vitro are completely different. In embryos, PrE is specified within 24h, whereas in vitro it takes 6 days. Thus it is difficult to see how these two timelines together with the different cell cycle lengths can be reliably compared. These limitations should be discussed in a revised manuscript.

We were aware of the difficulties inherent in this comparison and apologize if our statements were not sufficiently conditional. We have now added caveats to the Results section (line 328) and a revised discussion that explicitly discusses the difference in time lines (lines 404-409).

On line 111, we added a clarification that our in vitro model is a tool for deconstructing “transcriptional signatures” as this is precisely what we measure both in vivo and in vitro.

On line 322 we clarified that even though the time frames are different, we found a similar G1 signature trend in our in vitro dataset than the one found in the in vivo published dataset.

On line 337 we amended the sentence to clarify that some aspects of cell cycle regulation found in vitro could be also exploited in vivo.

On line 414 we have added a caveat line explaining that the use of G1 to drive increasing levels of commitment is a facet of cell fate choice both in vivo and in vitro, even though the time scales are different.

3) At least for the PrE differentiation conditions, the authors need to better emphasize the strong survival effects of the media and discuss the relative contributions of selective cell death and cell cycle changes for generating a homogeneously differentiated PrE culture.

As discussed above, we have now added discussion of the contribution of cell survival (lines 140-142, 162-168 and 342-347) and adapted our model to evaluate contributions of differences in survival and doubling times to differentiation (see Point 1 above).

4) The authors should address whether RACL is purely selective through changes to cell survival and cell cycle, or does it in addition trigger an increase in priming towards the Hhex-positive population before the acceleration of the cell cycle between days 2 and 3? It would be informative to compare transition rates towards the Hhex-high state between NACL and RACL to address this question

To address this point we compared transition rates towards the High Hhex state in the different media conditions, and found that there is an increase in priming towards the High Hhex population in days 2 and 3 of PrE differentiation (compared to NACL). We thank the reviewer for this comment and we now discuss this point (lines 223-225) and have added this to Figure 4 – Supplement 1D.

5) The authors should clarify their reasoning as to why cell cycle synchronization is relevant for the differentiation of a culture, or lineage priming (Lines 72, 73, 235). How does the absence or presence of cell cycle synchronization between daughter cells impact the composition of cell populations at a steady state or upon differentiation, as both long and short cell cycles appear to be equally well correlated?

This is a very good question and we can only speculate as to how cell synchronization impacts on population composition. We have added a discussion of this point (lines 393-403) and quote it here for convenience.

“If changes in proliferation are a response to differentiation cues, then synchronous inheritance could ensure that daughter cells retain an equivalent enhanced cell cycle state from their parents regardless of their position relative to the source of the signal. in vivo, where PrE differentiation occurs in a salt and pepper pattern in three dimensions, synchronization could enable cells to retain the memory of a signal regardless of their position and despite cell mixing. Moreover, if G1 represents the window of opportunity for cells to act upon signalling cues, we envision cell synchronization could represent inheritance of signalling opportunity, promoting similar durations of active transduction in clones of differentiating cells. If ERK is regulating both differentiation and cell cycle synchronization via G1, it is also possible that cell cycle synchronization is a consequence of ERK mediated alterations to the temporal window over which a cell responds to cytokine signalling”

6) The use of the term "differentiation" is somewhat confusing (,e.g. in line 136 "differentiation proceeds by providing an environment that favored the proliferation of a PrE-primed population"). As this is an initially heterogeneous cell population transiting to a more homogeneous, different state there may be a better way of describing this without terming it differentiation.

We have changed the phrasing on line 136, now corresponding to line 139. Current phrasing is as follows:

“This is the point in time when PrE clones became distinguishable from NEDiff cells in the time lapse video (Video 1) and suggested that culture conditions promoted endoderm specification by providing an environment that favoured the proliferation of a PrE-primed population.”

7) The data underlying the table in Figure S5C seem important as they address how state transitions shape the population structure in addition to the cell cycle properties of the different compartments. This should be discussed more extensively in a revised manuscript and perhaps moved to the main Results section. It would also be interesting to quantify residence times and rates of state transitions in units of time-1. These analyses could then be quantitatively compared to corresponding transitioning rates in the presence of PD03 (current Figure S8A, and lines 221 – 225)

We have moved Figure S5C to Figure 4E and discussed it further in the text (lines 213-216). We have also calculated transitions times using the division times from the dataset, and we added this to Figure 4 – Supplement 1C (and lines 217-220) for NACL and Figure 5 – Supplement 2A for PD03 (line 247). We observed the same trend for transition time than what we described for transition probability.

We have also calculated residence times for each state in NACL and PD03 (Table 4) and discussed it further in the text (Lines 221-223 for NACL and 250 for PD03). We calculated residence times based on generations and in hours.

Based on this information, it appears that the major impact of PD03 is to reduce the residence time in the High Hhex state while increasing the residence time in the Low Hhex state.

8) In Line 24 of the abstract the authors state that there is a lengthening of G1 in PrE-favouring conditions, but in the main body of the text it is the proportion between G1 and the rest of the cycle that is changing, (“Thus, G1 either remains unaffected or is marginally increased despite the decreasing cell cycle time (Figure 6A, Table S5)"). Similar claims about the lengthening of the G1 are made in the Discussion. The authors should ensure that the result is described consistently throughout the manuscript

While the claim in line 24 refers to PrE differentiation conditions, where we see a lengthening of G1 phase (Figure 6B), the line referring to G1 being marginally increased refers to Figure 6A, where G1 and division times are measured in steady state conditions (NACL, PD03 and 2i/LIF). We apologize if our claims were confusing and we have added a clarification in lines 297 and 318.

9) In Line 123 of the Results section, the authors state that: "The identity and distribution of PrE and NEDiff states were confirmed with GATA6 and NANOG staining (Figure S4A)." but the data presented by the authors are not sufficient to support this claim. Pairwise correlation between individual marker intensity needs to be calculated to prove this point with appropriate controls (for example with the use of the same primary antibody with two different secondary antibodies)

We confirmed the progressive acquisition of GATA6 cells and the loss of NANOG cells at different time points of PrE differentiation by quantifying multiple images from the immunostaining to Figure 2 —figure supplement 2B, together with a negative control corresponding to a sample without a primary antibody. The plots show individual datapoints corresponding to single cells, segmented based on DAPI staining.

To address this point we have also revised the text (line 127) to read “The identity and time evolution of PrE and NEDiff states were confirmed with GATA6 and NANOG staining (Figure 2 —figure supplement 2A, B)." We did this as we realized the word “distribution” may be misleading.

10) In Line 274 of the Results section, the authors state: "As we cannot combine the G1 reporter with Hhex lineage, we cannot assess whether it is the High Hhex or Low Hhex cells which present a longer G1." – I wonder why the authors cannot use here the same combination of the time-lapse plus immunostaining images as in the earlier experiments.

High and Low Hhex states are too dynamic to infer cell states based on antibody staining of ESC culture. If we stain with Hhex at the end of a FUCCI time lapse, there is no way to infer whether that High Hhex cell was also High 3 days prior to the staining. In the case of GATA6, we were labelling cells at the end of differentiation, and then inferring the nature of progenitors of these differentiated cells.

Our means of assigning dynamic identity in ESCs therefore required a different approach. Here, we know that fast dividing cells correspond to the Low Hhex, and slow dividing cells represent the High Hhex fraction. We therefore used division time to infer cell identity and found that fast dividing cells have longer G1 Ratio than slow cells (0.15 ± 0.055 vs 0.141 ± 0.043), suggesting Low Hhex cells have a longer G1, as indicated in the Results section, line 303-305.

11) The figure legend for figure S1 needs more detail so that it can be followed without cross-referencing with the main text and figure 1.

We have supplied detail.

References:

Argelaguet, Ricard, Stephen J. Clark, Hisham Mohammed, L. Carine Stapel, Christel Krueger, Chantriolnt-Andreas Kapourani, Ivan Imaz-Rosshandler, et al. 2019. ‘Multi-Omics Profiling of Mouse Gastrulation at Single-Cell Resolution’. *Nature* 576 (7787): 487–91. https://doi.org/10.1038/s41586-019-1825-8.

Hayashi, Katsuhiko, Hiroshi Ohta, Kazuki Kurimoto, Shinya Aramaki, and Mitinori Saitou. 2011. ‘Reconstitution of the Mouse Germ Cell Specification Pathway in Culture by Pluripotent Stem Cells.’ *Cell* 146 (4): 519–32. https://doi.org/10.1016/j.cell.2011.06.052.

Rugg-Gunn, Peter J. 2022. ‘Induction of Human Naïve Pluripotency Using Chemical ResettingChemical Resetting’. In *Human Naïve Pluripotent Stem Cells*, edited by Peter Rugg-Gunn, 29–37. Methods in Molecular Biology. New York, NY: Springer US. https://doi.org/10.1007/978-1-0716-1908-7_3.

Yasunaga, Masahiro, Shinsuke Tada, Satomi Torikai-Nishikawa, Yoko Nakano, Mitsuhiro Okada, Lars Martin Jakt, Satomi Nishikawa, Tsutomu Chiba, Takumi Era, and Shin-Ichi Nishikawa. 2005. ‘Induction and Monitoring of Definitive and Visceral Endoderm Differentiation of Mouse ES Cells’. *Nature Biotechnology* 23 (12): 1542–50. https://doi.org/10.1038/nbt1167.

Ying, Qi-Long, and Austin G Smith. 2003. ‘Defined Conditions for Neural Commitment and Differentiation’. *Methods in Enzymology* 365 (January): 327–41.